**Data Availability Statement:** All relevant data are within the manuscript and its Supporting

# Hemodynamic profiles by non-invasive monitoring of cardiac index and vascular tone in acute heart failure patients in the emergency department: External validation and clinical outcomes

**Nicholas Eric Harrison**[1,2]*, **Sarah Meram**[1], **Xiangrui Li**[3], **Morgan B. White**[2], **Sarah Henry**[1], **Sushane Gupta**[1], **Dongxiao Zhu**[4], **Peter Pang**[2], **Phillip Levy**[1]

1 Department of Emergency Medicine, Wayne State University School of Medicine, Detroit, Michigan, United States of America, 2 Department of Emergency Medicine, Indiana University School of Medicine, Indianapolis, Indiana, United States of America, 3 Division of Biostatistics, Washington University School of Medicine, St. Louis, Missouri, United States of America, 4 Department of Computer Science, Wayne State University College of Engineering, Detroit, Michigan, United States of America

* Harrisne@iu.edu

## Abstract

### Background

Non-invasive finger-cuff monitors measuring cardiac index and vascular tone (SVRI) classify emergency department (ED) patients with acute heart failure (AHF) into three otherwise-indistinguishable subgroups. Our goals were to validate these "hemodynamic profiles" in an external cohort and assess their association with clinical outcomes.

### Methods

AHF patients (n = 257) from five EDs were prospectively enrolled in the validation cohort (VC). Cardiac index and SVRI were measured with a ClearSight finger-cuff monitor (formerly NexFin, Edwards Lifesciences) as in a previous study (derivation cohort, DC, n = 127). A control cohort (CC, n = 127) of ED patients with sepsis was drawn from the same study as the DC. K-means cluster analysis previously derived two-dimensional (cardiac index and SVRI) hemodynamic profiles in the DC and CC (k = 3 profiles each). The VC was subgrouped *de novo* into three analogous profiles by unsupervised K-means consensus clustering. PERMANOVA tested whether VC profiles 1–3 differed from profiles 1–3 in the DC and CC, by multivariate group composition of cardiac index and vascular tone.

Profiles in the VC were compared by a primary outcome of 90-day mortality and a 30-day ranked composite secondary outcome (death, mechanical cardiac support, intubation, new/emergent dialysis, coronary intervention/surgery) as time-to-event (survival analysis) and binary events (odds ratio, OR). Descriptive statistics were used to compare profiles by two validated risk scores for the primary outcome, and one validated score for the secondary outcome.

Information files. The datasets are attached as CSV files, and code for reproduction of the primary statistical analyses in R are attached in the submission.

**Funding:** The ClearSight monitor's manufacturer, Edwards Lifesciences Corporation, provided funding for this study in the form of the Edwards Lifesciences Investigator-Initiated Grant. The funding was used for research assistant time and the monitors used in the study. Edwards Lifesciences Corporation also provided salary support for PL. The specific roles of these authors are articulated in the 'author contributions' section. The funders had no role in study design, data collection and analysis, decision to publish, or preparation of the manuscript.

**Competing interests:** I have read the journal's policy and the authors of this manuscript have the following competing interests: PL - Consultant: Apex Innovations, AstraZeneca, BMS, Mespere, Novartis, Cardionomics, Baim Institute, Ortho Clinical Diagnostics, Roche Diagnostics, Siemens, Hospital Quality Foundation. Research Support: American Heart Association, Beckman Coulter, Agency for Healthcare Research and Quality (AHRQ), Blue Cross Blue Shield of Michigan Foundation (BCBSMF), Emergency Medicine Foundation, Edwards Lifesciences, CardioSounds, Michigan Department of Health and Human Services, Michigan Health Endowment Fund, Patient Centered Outcome Research Institute, National Institutes of Health (NIH)/National Institute on Minority Health and Health Disparities, NIH/ National Heart Lung Blood Institute (NHLBI). Other: Past Chair of the Accreditation Oversight Committee for the American College of Cardiology (ACC) and a member of the ACC's National Cardiovascular Data Registry Oversight Committee. PL received salary support from funder Edwards Lifesciences Corporation. PP – In the last one year, PP has received Research Support: Roche, Beckman-Coulter, Siemens, Ortho-Diagnostics, Abbott, and AHRQ NEH – Research Support: BCBSMF, Indiana Clinical Translational Science Institute (CTSI) SM, PM, SH, SG, MBW, XL, DX report no conflicts of interest. There are no patents, products in development or marketed products associated with this research to declare. This does not alter our adherence to PLOS ONE policies on sharing data and materials.

## Results

The VC had median age 60 years (interquartile range {49–67}), and was 45% (n = 116) female. Multivariate profile composition by cardiac index and vascular tone differed significantly between VC profiles 1–3 and CC profiles 1–3 (p = 0.001, $R^2$ = 0.159). A difference was not detected between profiles in the VC vs. the DC (p = 0.59, $R^2$ = 0.016).

VC profile 3 had worse 90-day survival than profiles 1 or 2 (HR = 4.8, 95%CI 1.4–17.1). The ranked secondary outcome was more likely in profile 1 (OR = 10.0, 1.2–81.2) and profile 3 (12.8, 1.7–97.9) compared to profile 2. Diabetes prevalence and blood urea nitrogen were lower in the high-risk profile 3 (p<0.05). No significant differences between profiles were observed for other clinical variables or the 3 clinical risk scores.

## Conclusions

Hemodynamic profiles in ED patients with AHF, by non-invasive finger-cuff monitoring of cardiac index and vascular tone, were replicated *de novo* in an external cohort. Profiles showed significantly different risks of clinically-important adverse patient outcomes.

## Background

Acute heart failure (AHF) accounts for 1 million emergency department (ED) visits annually in the United States(US), 80% of which result in hospital admission [1,2]. AHF 30-day mortality overall (8–10% [3]) greatly exceeds the threshold of typical emergency physician (EP) risk tolerance (0.5–1%) [4], and neither EP gestalt for AHF mortality risk [5,6] nor clinical decision rules (CDRs) yet provide predictive value sufficient [1,7] to meet such low risk thresholds. Consequently, half of ED to hospital admissions for AHF involve low-risk patients for whom admission may not be necessary [1,2,8–10], and recent Society for Academic Emergency Medicine (SAEM) and Heart Failure Society of America(HFSA) guidelines [1] stress the importance of developing new AHF risk markers in the ED. A particular need exists for novel markers which identify low-risk AHF presentations by way of capturing the high-degree of physiologic and clinical heterogeneity between AHF patients, given relatively more established predictors of high-risk [1,11] and the high baseline ED admission rate.

Classification by hemodynamic profile is one of the oldest approaches to subgrouping the high clinical heterogeneity present among AHF patients, given that hemodynamic derangements are critical defining features of AHF pathophysiology. Hemodynamic parameters like cardiac index, vascular tone (systemic vascular resistance index {SVRI}), heart rate, blood pressure (BP), and others reflect some of the greatest physiologic heterogeneity among AHF patients [12–18], and consideration of heart rate and BP are prominent features of contemporary ED AHF evaluation [14]. Cardiac index and vascular tone play an outsized role in AHF pathophysiology [14,15], yet are not generally able to be assessed in the ED. Gold standard measurement by pulmonary artery catheterization (PAC) requires specialist expertise, is highly invasive, and is employed in only 1% of contemporary AHF hospitalizations [19]. The Forrester classification of "wet-dry/warm-cold" on physical exam [20,21] is a non-invasive method for assessing cardiac index and vascular tone in AHF [1,22,23], but limited in clinical utility given a subjective nature and poor reliability, with interrater agreement of just 64% (kappa = 0.28) in ED patients [24].

Recently, a non-invasive monitor providing continuous estimation of cardiac index and vascular tone[18] was described in an ED-based retrospective study of the PREMIUM (Prognostic Hemodynamic Profiling in the Acutely Ill Emergency Department Patient) registry by Nowak et. al. ClearSight (formerly "NexFin", as it was known in this prior study; Edwards Life-sciences, Irvine, California) is an FDA-approved finger-cuff monitor which measures continuous blood pressures and pulse rates at both the radial and digital arteries. Finger-cuff monitors are attractive for ED profiling of patients by cardiac index and vascular tone, because they are non-invasive like the Forrester classification, yet provide the clinician with reproducible, continuous, and objective measurements like a PAC. Nowak et al. derived 2-dimensional hemodynamic profiles by cardiac index and vascular tone (SVRI) from the finger cuff measurements in these ED AHF patients. Profiling by these two hemodynamic variables is the same physiological construct underlying the Forrester classification, but with objective measurements guiding classification rather than highly-subjective [24] physical examination. Namely, profiling by cardiac index and vascular tone reflects that the ventricular-vascular relationship is naturally discordant, since maintenance of blood pressure requires any decrease in cardiac index to be buffered by an equal and opposite increase in vascular tone and visa-versa (i.e. Mean arterial BP = cardiac index x SVRI) [14]. Importantly, disruption of the ideal ventricular-vascular relationship, such as by nitrate metabolism, myocardial changes, neurohormonal effects, and other factors, is a key component of HF physiology [5,11,15,16,25–30].

Three distinct ED AHF hemodynamic profiles by cardiac index and vascular tone were described by Nowak et al. which, critically, were not otherwise identifiable by clinical characteristics [18]. This 3-profile system observed reflected a surprisingly large degree of previously uncaptured physiologic heterogeneity [18], spanning very low and very high values in both the cardiac index and vascular tone dimensions. The prior study had 3 primary limitations identified as needing further research by the authors: 1. Finger-cuff monitors hemodynamic monitoring in AHF is novel, and the ability to reproduce this 3-profile classification needed replication in an external sample (external validation). 2. Patients were not diagnostically adjudicated as having AHF, and it is unknown whether profiling on this monitor is specific to AHF versus other diagnoses in the ED 3. The study was not powered to detect prognostic difference between profiles, and thus clinical significance of these profiles for a goal of risk-stratification is unknown.

In the current study, we sought to address these limitations. We hypothesized that 1. De novo cluster analysis of a new prospective sample of AHF patients (validation cohort, VC) would produce cardiac index/vascular tone hemodynamic profiles matching the derivation cohort(DC) [18]; 2. VC profiles would not match profiling of patients in PREMIUM with sepsis (control cohort, CC) or those with non-cardiac dyspnea (CC2), and 3. Between-profile differences in the VC would exist for both 90-day all-cause mortality (primary outcome) and a 30-day ranked composite of adverse events [31,32] (secondary outcome).

## Methods

CLEAR-AHF was a multicenter prospective observational study approved as minimal-risk research by the Wayne State University (WSU) and Indiana University (IU) institutional reviews boards. The primary aim of the study was to create a multi-institutional registry of hemodynamic data in ED AHF patients, given the lack of other methods for measuring important hemodynamic parameters like cardiac index and SVRI (vascular tone) in the emergency department. A secondary aim was to validate the hemodynamic profiles derived in the pilot study [18] and determine whether these profiles were associated with clinically-important patient outcomes. Written consent was obtained from all participants. This manuscript was written to comply with the STROBE guidelines for cohort studies [33].

## Study setting and participants

Patients ≥18 years old presenting to the ED were screened for enrollment 24 hours per day at five EDs in the US. Annual ED volumes range from approximately 80,000–100,000 patient visits per site. Enrollment occurred from July 2017—March 2019.

Included patients had EP suspicion of AHF and at least one of the following: dyspnea at rest or exertion, signs of AHF on chest radiograph (CXR), and/or NT-proBNP>300 pg/ml. Patients with dyspnea primarily due to other causes (by EP diagnosis), temperature >38.5˚C or suspected sepsis, acute ST-elevation myocardial infarction (STEMI), pregnant women, prisoners, and those without an ejection fraction (EF) recorded within 12 months, were excluded. Two study authors with extensive AHF research experience (PP and PL), blinded to one another, performed case review to adjudicate the ED diagnosis of AHF. Disagreements were decided by discussion, and patients without diagnostically-adjudicated AHF were excluded.

## Study protocol

Patients were fitted with a ClearSight finger-cuff monitor and continuous measurements recorded for 3–6 hours. Manufacturer reference range values for cardiac index and SVRI are 2.5–4.0 L/min/m2 and 1970–2390 dynes-sec/cm5/m2, respectively. Clinicians and patients were blinded to device measurements by an opaque sheet on the monitor screen to prevent bias in management decisions. Patients were excluded if they could not begin monitoring in the ED. The median time from initial IV loop diuretic to initiation of hemodynamic monitoring in the sample was 98 minutes (IQR: 29–167), and the first recorded value for cardiac index and SVRI were used for the analysis.

Research assistants used standardized data sheets to record patient demographics, medications, medical history, vital signs, clinical tests, ED treatments, ED disposition, and hospital course. Data were obtained through patient phone interview, the electronic medical record (EMR), and the reports of treating physicians. Data were recorded in REDCap (Research Electronic Data Capture; http://project-redcap.org/).

## Measures

**Primary measure.**    The primary measure of interest was each patient's 3-level categorical hemodynamic profile, as derived in the DC [18]. Each profile is a subgroup of the multivariate distribution in two dimensions: cardiac index and vascular tone (SVRI). The first available cardiac index and vascular tone, obtained simultaneously, was used for profiling (see Analysis, Validation Parts 1–2). Finger-cuff hemodynamic monitors have >90% correlation to invasive arterial blood pressure monitoring [34], and estimate invasive cardiac index with a ~30% margin of error [18,35]. While the level of accuracy of this estimate does not imply interchangeability with invasive hemodynamic monitors [27], it is nevertheless accurate enough to be useful for initial assessment before an invasive monitoring can begin in the intensive care unit [27]. Since invasive hemodynamic monitoring is only performed in 1% of contemporary AHF hospitalizations [19] and is outside the scope of practice of EPs, a non-invasive estimate such as from a finger-cuff monitor is the only feasible measure for cardiac index in the ED. For more details on finger-cuff hemodynamic monitors, see the publication for the DC [18] and other prior literature [27].

Cluster analysis methods to subgroup patients by cardiac index and vascular tone into one of three hemodynamic profiles are described below for the VC (see Validation Part 1) and in the prior publication [18] for the DC. The CC hemodynamic profiles were obtained in an unpublished analysis by the same methods and at the same time as the DC [18], from a

concurrently enrolled group of septic patients in the same registry (PREMIUM) as the DC's AHF patients.

Hemodynamic profiling by cardiac index and vascular tone was performed *de novo* in the VC (see Validation Part 1), to test if profiling of the VC replicated the profiles of AHF patients in the DC [18] (hypothesis 1) and differed from the profiles of non-AHF patients in the CC (hypothesis 2). For consistency, profile numbering 1 through 3 in the VC was set to match the corresponding profiles 1–3 in the DC and CC.

**Secondary measures.** Other measures included numerous clinical variables: demographics, vital signs, laboratory tests performed in the ED, chest x-ray (CXR) and electrocardiogram (ECG) findings. Three validated CDRs for AHF-risk stratification in the ED were also calculated as well to place our study in context. The Emergency Heart Failure Mortality Risk Grade (EHMRG) and Get With The Guidelines HF Risk Score (GWTG-HF) have were derived [5,36] and externally validated [37–39] to predict short-term AHF mortality. The STRATIFY risk score was derived [1] and validated [32] in a US ED population to predict a 30-day ranked composite of clinically-important AHF adverse events (study secondary outcome, least to most severe): 1. invasive cardiac procedure or acute coronary syndrome (IP/ACS), 2. new or emergent dialysis (NED), 3. intubation, 4. mechanical cardiac support or transplant (MCS/T), 5. death or cardiopulmonary resuscitation (D/CPR). The GWTG-HF risk score has been shown to have both short-term [36] and long-term [38,39] prognostic value for AHF mortality. EHMRG was previously demonstrated to outperform EP gestalt [5,37] for short-term AHF mortality.

## Outcomes

The primary clinical outcome was mortality within 90 days of ED presentation. The secondary outcome was the hierarchical 30-day composite used in STRATIFY, described above (Secondary Measures). All events in the secondary outcome hierarchy were recorded for each patient. Multiple occurrences of the same event were treated as a single event for analysis purposes. The primary and secondary outcomes were recorded as both days-to-event (survival) and binary variables.

Study coordinators collected outcome information by EMR follow-up and telephone interview at 30 days, 90 days, 180 days, and 1 year. Both institutions have large hospital networks (4 hospitals WSU,14 hospitals IU) sharing EMR data with inpatient and outpatient services. Both also participate in state-wide healthcare information exchanges (HIEs). HIE data was used to augment patient telephone follow-up of adverse events occurring at outside hospital systems. Follow-up through the HIEs, telephone interviews, and local EMRs resulted in 100% confirmation of survival vs. death at 90 days. Records for each outcome were independently queried by two or more data abstractors blinded to the analysis.

## Statistical analysis

All analyses were conducted in the R statistical programming language (http://www.r-project. org, The R Foundation, v3.6.1). Cluster analysis for the VC was performed in the ConsensusClusterPlus R package [40]. 3 sensitivity analyses were performed, with details in Supplement S1 File. Fully-runnable code and deidentified minimal data sets are included in Supplement S2–S5 Files.

**Validation part 1: De novo identification of hemodynamic profiles in the validation cohort.** All patients were clustered *de novo* to ensure that the cluster algorithm assignment of hemodynamic profiles to VC patients was naïve to all patient data besides cardiac index and vascular tone, and naïve to the data and profiling in the DC [18]. We reasoned that this would

help test the criterion validity of the marker of interest. First, if hemodynamic profile on the finger-cuff device represents a reproducible feature of AHF patient physiology, then profiling in the DC and VC should not appreciably differ by multivariate distribution of cardiac index and vascular tone. If they did differ, it would suggest that profiling of the DC represented random statistical noise and/or overfitting of the data rather than a reproducible physiologic feature. Second, we hypothesized the VC profiles would differ in cardiac index and vascular tone from the profiling of ED patients with sepsis in the CC. Namely, if the hemodynamic profiles in the VC represent a construct specific to AHF hemodynamics, the VC profiles should be distinguishable from 3-category profiling of a condition with markedly different hemodynamics like sepsis. Septic patients in the CC were enrolled at the same time and at the same centers as the AHF patients in the DC, as part of the PREMIUM registry.

**Validation part 2: Unsupervised machine learning cluster analysis method for profiling of the validation cohort.** Standard K-means clustering was used in the DC to derive the original hemodynamic profiles [18] but has two major weaknesses: 1. the data analyst must decide before clustering how many clusters (k) to subgroup the data by, and 2. clustering of the DC was performed without internal validation (i.e. using the entire cohort). Both weaknesses could potentially lead to overfitting of each class/profile to the data set. For the *de novo* profiling of the VC we used a machine learning tool called consensus clustering [40] to identify two-dimensional patient clusters. As in standard k-means clustering, the input is multivariate data (cardiac index and vascular tone) for each individual in the study. In consensus clustering there is no assumption of the ideal number of clusters, and internal validation to assess cluster stability is performed unsupervised through random resampling and introduction of random data perturbations [40]. The result a reduction in the chance of spurious class discovery due to overfitting of the dataset and less reliance on analyst-provided assumptions. Supplement S1 File gives further methodological details on consensus clustering and how it differs from k-means. While the DC study's authors subjectively chose k = 3 profiles for patient classification, we allowed the learning machine to split the data into anywhere between 1–10 unique hemodynamic profiles. Nonetheless, 3–4 groups maximized the internally-validated consensus scoring based on examining consensus score dendrograms and the elbow plot of change in cumulative distribution function for each subsequent level k 1–10 (supplement S1 File).

Comparisons of VC profiling with the DC and CC profiles were first made qualitatively by superimposing scatter plots (cardiac index vs. vascular tone) of the cohorts, along with their hemodynamic profiles. Quantitative analyses were performed by PERMANOVA, in which VC profiling was compared to the DC and CC profiles for multivariable similarity of group composition by cardiac index and vascular tone. Cardiac index and vascular tone were the independent variables of the PERMANOVA models, cohort as the dependent variable, with permutations (n = 999) blocked by hemodynamic profile (1–3).

**Validation part 3: Comparison of hemodynamic profiles in the validation cohort with the derivation and control cohorts.** Comparisons of VC profiling with the DC and CC profiles were first made qualitatively by superimposing scatter plots (cardiac index vs. vascular tone) of the cohorts, along with their hemodynamic profiles. Quantitative analyses were performed by PERMANOVA, in which VC profiling was compared to the DC and CC profiles for multivariable similarity of group composition by cardiac index and vascular tone. In each of two PERMANOVA models (VC vs. DC, VC vs. CC), observations/patients in the comparator cohort (VC) were combined with the reference cohort (DC or CC) into a single dataset with the following variables for all observations: cardiac index, vascular tone, hemodynamic profile number assigned during clustering, and cohort name. Cardiac index and vascular tone were the independent variables of the PERMANOVA models, cohort as the dependent variable, with permutations (n = 999) blocked by hemodynamic profile (1–3). We chose α = 0.30

as the level of statistical significance for the hypotheses that cardiac index and vascular tone within each profile differed between cohorts ($H_1$: VC vs. DC, $H_2$: VC vs. CC). $R^2$ for each model were reported, representing the proportion of between-cohort variance in cardiac index and vascular tone within each profile 1–3.

**Comparison of validation cohort profiles by clinical features.** In the DC [18], no significant ($\alpha = 0.05$) difference in clinical variables was detected, suggesting that the hemodynamic profiles were not readily explained by other common clinical markers and therefore more likely to be a novel marker unto themselves. In the VC, patients were compared by profile for each clinical variable and CDR similarly. Continuous variables were compared with the non-parametric Kruskall-Wallis test or parametric ANOVA. Categorical variables were assessed with the chi-square test.

**Comparison of validation cohort profiles with clinical study outcomes.** The primary and secondary outcomes were assessed first by survival analysis. Kaplan-Meier curves stratified by hemodynamic profile were produced and then compared with the log-rank test. Survival for the secondary outcome was defined as freedom from any fatal or non-fatal adverse event in the hierarchy. Odds ratios (OR) with 95% confidence intervals (CI) were calculated for between-profile comparisons of the categorical parameterization of the primary (binary) and secondary (ordinal/ranked) outcomes.

## Results

### Participants and description of the validation cohort

Of 351 patients screened against inclusion criteria for the VC and who consented for hemodynamic monitoring, 17 did not begin monitoring in the ED, 4 did not have a recorded EF, and 78 did not have AHF on diagnostic adjudication (Fig 1). The 257 remaining patients had a median age of 60 years (interquartile range{IQR} 49–67) and 45% were female.

Table 1 presents VC patient demographics, medical history, outpatient medications, initial vital signs, labs, interventions, clinical testing, risk scores, and initial values for cardiac index and vascular tone. 25% required supplemental oxygen, and 9% received a critical care intervention. 47% had HF with reduced EF (HFrEF), 79% of whom were adherent to guideline directed medical therapy. Median (IQR) for cardiac index and vascular tone were 2.40 L/min/m$^2$ (2.00–3.08) and 3196 dynes-sec/cm5/m2 (2578–3919), respectively. Characteristics of the DC have been described previously [18].

### Hemodynamic profiling in the validation cohort

The consensus clustering algorithm was performed in the VC based on cardiac index and vascular tone (SVRI). Inspection of the consensus dendrograms [40] (Supplement S1 File) for each k clusters 1–10 showed the cleanest divisions to occur when the data was divided into k = 3–4 groups. Inspection of the delta area change in consensus score CDF [40] for K 1–10 (elbow plot, Supplement S1 File) show minimal improvement in area under the CDF curve for k>3. Taken together, these suggest that further divisions of the data (i.e. more profiles / increasing K) beyond k = 3 resulted primarily in data sorting at random rather than improved classification [40] by cardiac index and vascular tone in the VC. Consequently, the clustering and internally-validation performed by the consensus algorithm at k = 3 were designated hemodynamic profile 1 (n = 68), 2 (n = 69), and 3 (n = 120) in the VC.

Cardiac index, vascular tone, and clinical characteristics by profile are presented in Table 1. Profiles 1–3 resembled each other (p>0.05) for all clinical variables except BUN and history of diabetes, with each being highest in profile 1 and lowest in profile 3 (Table 1). Vascular tone and profiling overall did not appear to be a simple function of blood pressure, with no

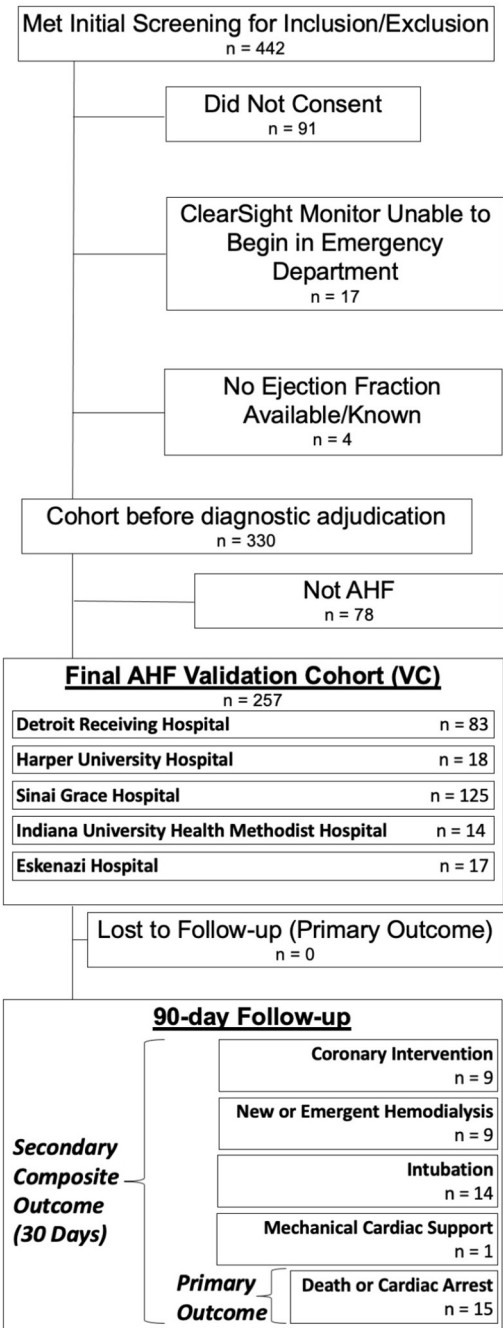

**Fig 1. Validation cohort flow diagram.** AHF = Acute Heart Failure.

significant differences in SBP (p = 0.461) or DBP (p = 0.933) between profiles, which aligns with our prior published work on this device in an ED population of AHF patients [14] showing only moderate to low correlation of SVRI to device-estimated DBP (r = 0.587), SBP (r = 0.324), and mean arterial pressure ({MAP}, r = 0.479).

The 3 validated clinical risk scores (EHMRG, GWTG-HF, and STRATIFY) had no statistically significant difference between profiles (Table 1).

**Table 1. Clinical characteristics of patients in the validation cohort.**

| Patient Characteristic | % (count) or median (IQR) | | | | |
|---|---|---|---|---|---|
| | All Patients {n = 257} | Hemodynamic Profile 1 {n = 68} | Hemodynamic Profile 2 {n = 69} | Hemodynamic Profile 3 {n = 120} | p-value |
| *Demographics, Medical History, and Outpatient Medications* | | | | | |
| Age (years) | 60 (49–67) | 59 (46–65) | 62 (52–69) | 59 (51–66) | 0.080 |
| Female | 45% (116) | 43% (29) | 49% (34) | 44% (53) | 0.707 |
| Body Mass Index (kg/m$^2$) | 32 (26–39) | 33 (28–40) | 31 (25–39) | 31 (25–38) | 0.366 |
| African American Race | 89% (229) | 85% (58) | 90% (62) | 91% (109) | 0.490 |
| EMS transport to ED | 35% (89) | 38% (26) | 33% (23) | 33% (40) | 0.767 |
| Hypertension | 93% (240) | 96% (65) | 96% (66) | 91% (109) | 0.305 |
| Diabetes | 49% (126) | 63% (43) | 46% (32) | 43% (51) | 0.021*** |
| Chronic Kidney Disease | 38% (97) | 47% (32) | 33% (23) | 35% (42) | 0.177 |
| Dialysis | 7% (19) | 10% (7) | 7% (5) | 6% (7) | 0.531 |
| Pulmonary Hypertension | 29% (74) | 29% (20) | 33% (23) | 26% (31) | 0.544 |
| Valvular Disease | 35% (91) | 35% (24) | 46% (32) | 29% (35) | 0.059 |
| COPD | 38% (98) | 35% (24) | 39% (27) | 39% (47) | 0.854 |
| Active Cancer | 3% (7) | 2% (1) | 6% (4) | 2% (2) | 0.186 |
| Heart Failure | 93% (238) | 90% (61) | 93% (64) | 94% (113) | 0.531 |
| HFrEF, on GDMT | 37% (95) | 28% (19) | 44% (30) | 38% (46) | 0.155 |
| HFrEF, not on GDMT | 10% (25) | 15% (10) | 4% (3) | 10% (12) | 0.122 |
| HFpEF | 53% (137) | 57% (39) | 52% (36) | 52% (62) | 0.736 |
| Ejection Fraction (%) | 40 (25–59) | 45 (25–60) | 40 (20–50) | 40 (25–60) | 0.127 |
| ACEi or ARB | 48% (124) | 46% (31) | 55% (38) | 46% (55) | 0.415 |
| Beta Blocker | 70% (180) | 72% (49) | 70% (48) | 69% (83) | 0.913 |
| Loop Diuretic | 61% (157) | 62% (42) | 54% (37) | 65% (78) | 0.301 |
| Metolazone | 2% (5) | 0% (0) | 3% (2) | 3% (3) | 0.392 |
| Antiarrhythmic | 6% (15) | 4% (3) | 4% (3) | 8% (9) | 0.567 |
| *ED Initial Vitals, Labs, and Interventions* | | | | | |
| Systolic Blood Pressure (mmHg) | 154 (132–178) | 160 (139–177) | 154 (129–175) | 150 (129–180) | 0.461 |
| Diastolic Blood Pressure (mmHg) | 91 (80–105) | 91 (77–111) | 91 (80–107) | 92 (81–104) | 0.933 |
| SpO$_2$ (%) | 97 (95–99) | 98 (96–99) | 98 (96–99) | 97 (95–98) | 0.258 |
| Respiratory Rate | 20 (18–22) | 20 (18–22) | 18 (18–22) | 20 (18–24) | 0.170 |
| Heart Rate | 92 (80–105) | 94 (84–105) | 89 (79–101) | 90 (81–106) | 0.428 |
| Sodium (mmol/L) | 139 (137–142) | 138 (136–141) | 140 (138–142) | 140 (137–142) | 0.107 |
| Potassium (mmol/L) | 4.1 (3.8–4.5) | 4.1 (3.8–4.4) | 4.1 (3.9–4.7) | 4 (3.6–4.5) | 0.265 |
| Blood Urea Nitrogen (mg/dL) | 21 (15–31) | 25 (16–40) | 20 (16–30) | 19 (15–27) | 0.033*** |
| eGFR (mL/min/1.73m$^2$) | 63 (37–86) | 55 (27–79) | 61 (37–83) | 66 (42–89) | 0.129 |
| Troponin I (ng/mL) | 0.032 (0–0.07) | 0 (0–0.096) | 0.04 (0–0.079) | 0.032 (0–0.065) | 0.545 |
| Troponin Positive | 44% (113) | 38% (26) | 48% (33) | 46% (55) | 0.493 |
| BNP (pg/mL) | 1067 (414–2055) | 1524 (562–2337) | 1113 (443–2575) | 770 (376–1967) | 0.051 |
| Supplemental O$_2$ | 25% (65) | 28% (19) | 19% (13) | 28% (33) | 0.353 |
| IV Vasoactive, Inotrope, or PPV | 9% (23) | 9% (6) | 7% (8) | 13% (9) | 0.335 |
| *Electrocardiogram (ECG) and Chest X-Ray (CXR)* | | | | | |
| Wide QRS | 16% (42) | 10% (7) | 25% (17) | 15% (18) | 0.106 |
| A-fib or A-flutter | 12% (32) | 6% (4) | 15% (10) | 15% (18) | 0.362 |
| Q Waves | 9% (22) | 3% (2) | 15% (10) | 8% (10) | 0.079 |
| Normal ECG | 34% (87) | 40% (27) | 29% (20) | 33% (40) | 0.199 |
| Alveolar Edema | 19% (48) | 22% (15) | 16% (11) | 18% (22) | 0.542 |

*(Continued)*

**Table 1.** (Continued)

| Patient Characteristic | % (count) or median (IQR) | | | | p-value |
|---|---|---|---|---|---|
| | All Patients {n = 257} | Hemodynamic Profile 1 {n = 68} | Hemodynamic Profile 2 {n = 69} | Hemodynamic Profile 3 {n = 120} | |
| Interstitial Edema | 13% (34) | 19% (13) | 12% (8) | 11% (13) | 0.201 |
| Cardiomegaly | 85% (218) | 88% (60) | 84% (58) | 83% (100) | 0.647 |
| Hyperinflated | 5% (13) | 3% (2) | 6% (4) | 6% (7) | 0.661 |
| Normal CXR | 5% (14) | 3% (2) | 6% (4) | 7% (8) | 0.513 |
| *AHF Clinical Decision Rule Scores, and Earliest ED Finger-cuff Hemodynamic Measurements* | | | | | |
| STRATIFY | 198 (180–226) | 209 (180–252) | 194 (178–219) | 197 (181–223) | 0.317 |
| GWTG-HF Risk Score | 31 (27–37) | 31 (26–38) | 33 (27–38) | 31 (28–36) | 0.737 |
| EHMRG† | -17 (-59.4–28.9) | -26 (-70.8–28) | -11.5 (-44.3–41) | -15.6 (-59.6–28.4) | 0.309 |
| First Cardiac Index (L/min/m²) | 2.4 (2.00–3.08) | 3.71 (3.35–4.12) | 1.81 (1.64–1.97) | 2.4 (2.18–2.65) | p<0.001*** |
| First Vascular Tone {SVRI} (dynes-sec/cm⁵/m²) | 3196 (2578–3919) | 2240 (1745–2572) | 4479 (4032–5260) | 3180 (2801–3539) | p<0.001*** |

**Table 1 Legend**—EMS = Emergency medical services; ED = Emergency department; COPD = Chronic obstructive pulmonary disease; HFrEF = Heart failure with reduced ejection fraction; HFpEF = HF with preserved EF; ACEi = Angiotensin converting enzyme inhibitor; ARB = Angiotensin receptor blocker; SpO2 = Oxygen saturation; eGFR = estimated glomerular filtration rate; BNP = Brain natriuretic peptide; O$_2$ = oxygen; GWTG-HF = get-with-the-guidelines heart failure; EHMRG = Emergency Heart Failure Mortality Risk Grade.

†EHMRG as calculated in Table 1 excludes patients who were dialysis dependent, as the EHMRG was derived and validated in a population excluding such patients. The total cohort and profile sizes without dialysis history were total cohort n = 238, profile 1 n = 61, profile 2 n = 64, profile 3 n = 113. None of the patients with dialysis history died within 90 days.

## Comparison of hemodynamic profile composition in the validation cohort to the derivation and control cohorts

Fig 2 shows cardiac index and vascular tone for the 3 profiles in the VC alone (panel A) and compared to the DC [18] (panel B) and CC (panel C) patients' hemodynamic profiles. Multi-variate statistical comparison by PERMANOVA did not show the profiles in the DC [18] and VC to be significantly different at the prespecified $\alpha$ = 0.3 threshold (Fig 2B, p = 0.59, R$^2$ = 0.016). A significant difference (PERMANOVA p = 0.001, R$^2$ = 0.159) in cardiac index and vascular tone was present between the VC and CC profiles (Fig 2C).

## Clinical outcomes by hemodynamic profile in the validation cohort

Outcomes rates by profile in the VC are presented in Table 2. 89% of patients were admitted to the hospital or an observation unit, 6% died within 90 days (primary outcome), and 7% experienced ≥1 fatal or non-fatal 30-day adverse event in the composite secondary outcome (Table 2). 90-day mortality (primary outcome) was significantly more likely (OR = 5.0, 95%CI 1.4–18.0) in Profile 3 compared to Profiles 1 or 2 (Table 2). Profile 3 also had shorter time to death than 1 or 2 (Table 2), including every death within 30 (p = 0.049) and 60 days (p<0.001).

Comparison of 30-day events by profile are presented in Fig 3A and Table 2. Fig 3B shows rates for ED critical care interventions, unstable vital signs, dispositions, and loop diuretic administration all of which were similar between profiles (p>0.05). A 30-day fatal or non-fatal adverse event (secondary outcome) occurred in 16% of profile 3 patients, 13% profile 1, and 1% of profile 2 (p = 0.008). Median event severity/rank in profile 2 was lower compared to 1 or 3 (p = 0.003, Table 2). The likelihood of any 30-day adverse event (Table 2) was higher in Profile 3 vs. Profile 2 (OR = 12.8, 95%CI: 1.7–97.9) and Profile 1 vs. 2 (OR = 10.0, 95%CI: 1.2–81.2), but similar in Profile 3 vs. Profile 1 (OR = 1.28, 95%CI 0.5–3.0).

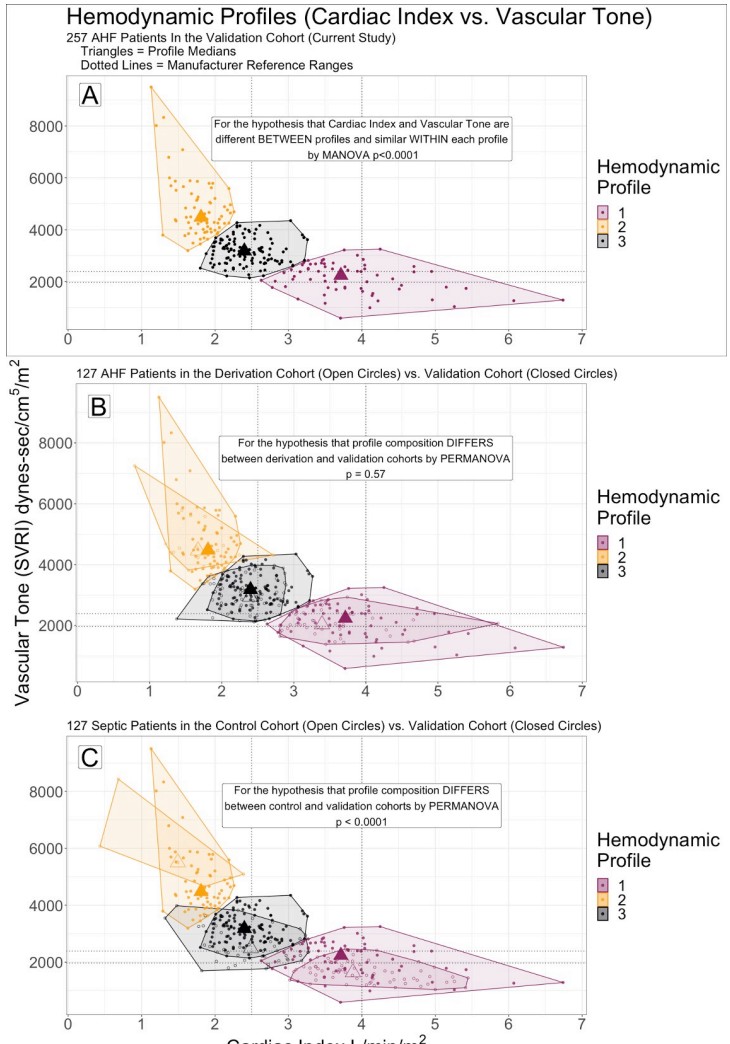

**Fig 2. Hemodynamic profiles by cardiac index vs. vascular tone (systemic vascular resistance index, SVRI).** Profiles were numbered similarly to facilitate between cohort comparisons: 1 (purple)—lowest SVRI and highest cardiac index, 2 (gold)—highest SVRI and lowest cardiac index, 3 (black) cardiac index and SVRI between profiles 1 and 2. (A) Profiling in the validation cohort (VC) alone. (B) The VC patients and their profiles are overlayed with the derivation cohort (DC). Patients in the DC had acute heart failure and were monitored in the emergency department like the VC, but were enrolled in a prior study (external cohort). Few patients classified in a particular profile in the VC would have been classified differently in the DC. (C) The VC overlayed with the control cohort (CC). The CC included patients enrolled in the same study as the DC, but who had sepsis rather than AHF. Profiling in the CC differed from VC, with several VC patients who would have been classified in a different profile by profiling of the CC (and visa versa).

Fig 4 presents Kaplan-Meier survival for mortality (4A) and the secondary outcome(4B) stratified by profile and compared at each of 30, 60, and 90 days (all log-rank p<0.05, both outcomes). Time-to-death was worst in profile 3 (90-day hazard ratio {HR} = 4.83, 95%CI 1.36– 17.1), while time to any event (90-day HR = 0.36, 95%CI: 0.15–0.85) was best in profile 2.

## Sensitivity analyses

On principle components analysis of the VC hemodynamic profiles, using all clinical variables collected (Table 1) other than cardiac index and vascular tone, we found that $\geq 7$ principal components were required to explain $\geq 75\%$ of between-profile variance.

**Table 2. Study outcomes overall and by hemodynamic profile.**

| Outcome | % (count) or mean (SD) | | | | |
|---|---|---|---|---|---|
| | All Patients {n = 257} | Hemodynamic Profile 1 {n = 68} | Hemodynamic Profile 2 {n = 69} | Hemodynamic Profile 3 {n = 120} | p-value |
| *Mortality (Primary Outcome)* | | | | | |
| 30-Day | 2% (6) | 0% (0) | 0% (0) | 4% (5) | p = 0.049*** |
| 60-Day | 4% (11) | 0% (0) | 0% (0) | 9% (11) | p<0.001*** |
| 90-Day | 6% (15) | 3% (2) | 1% (1) | 10% (12) | p = 0.027*** |
| Time to Death (days) | 87.3 (12.6) | 89.7 (1.7) | 89.7 (2.2) | 84.6 (17.9) | p = 0.016*** |
| *30-Day Ranked (0–5) Composite of Adverse Events (Secondary Outcome)*—adapted from Collins 2015[31] | | | | | |
| 0  Event-free at 30 days | 93% (239) | 87% (59) | 99% (68) | 84% (101) | p = 0.008*** |
| 1  Invasive Cardiac Procedure | 5% (12) | 6% (4) | 1% (1) | 6% (7) | p = 0.187 |
| 2  New or Emergent Dialysis | 3% (8) | 3% (2) | 0% (0) | 5% (6) | p = 0.300 |
| 3  Intubation | 5% (13) | 6% (4) | 0% (0) | 8% (9) | p = 0.181 |
| 4  Mechanical Cardiac Support or Transplant | < 1% (1) | 0% (0) | 1% (1) | 0% (0) | p = 0.601 |
| 5  Death or Cardiopulmonary Resuscitation | 2% (6) | 0% (0) | 0% (0) | 4% (5) | p = 0.049*** |
| Mean Rank of Worst Event in 30 days (0–5) | 0.30 (0.98) | 0.28 (0.79) | 0.06 (0.48) | 0.46 (1.24) | p = 0.003*** |
| *Other Outcomes*—all at 30-day follow-up unless otherwise indicated | | | | | |
| Discharged from Emergency Department | 11% (28) | 10% (7) | 14% (10) | 9% (11) | p = 0.268 |
| Admission or Transfer to ICU | 18% (45) | 22% (15) | 15% (10) | 17% (20) | p = 0.480 |
| Index Hospitalization Length of Stay (days) | 4 (4) | 5 (5) | 4 (3) | 4 (4) | p = 0.935 |
| ED Revisits for AHF | 18% (46) | 18% (12) | 15% (10) | 20% (24) | p = 0.635 |
| ED Revisits, All-Cause | 26% (68) | 31% (21) | 22% (15) | 27% (32) | p = 0.478 |
| AHF Readmissions | 16% (41) | 16% (11) | 13% (9) | 18% (21) | p = 0.722 |
| All-Cause Readmissions | 22% (56) | 27% (18) | 17% (12) | 22% (26) | p = 0.436 |
| ICU or Death | 18% (46) | 22% (15) | 15% (10) | 18% (21) | p = 0.507 |

**Table 2 Legend**—ICU = Intensive care unit; ED = Emergency Department; AHF = Acute heart failure.

116/257 (45.1%) patients met our specified criteria (Supplement S1 File—Sensitivity analysis 2) for a "clear indication" for hospital admission in the ED. After excluding these patients, just 12.1% of those remaining (17/141) were discharged from the ED, but 56.7% (80/141) were hemodynamic Profile 1 or 2 (low risk for 90-day mortality) and 30.0% (42/141) were Profile 2 (low risk for any 30-day adverse event). If Profile 2 vs. Profile 1 or 3 was used as a discharge criterion in these 141 patients without a clear indication for hospital admission, significantly more patients (p<0.001) would have been discharged from the ED compared to the actual discharge rate, without any missed deaths (100% negative predictive value for Profile 2).

CC2 (patients with non-cardiac dyspnea) had p = 0.001 difference in profiling compared to AHF-adjudicated patients in the VC by PERMANOVA.

## Discussion

In this multicenter prospective observational study, we report 3 main objectives and their findings for what is to our knowledge the first time: 1. We externally validated non-invasive hemodynamic profiling of ED AHF patients by cardiac index and vascular with a finger-cuff monitor, 2. We show that this profiling was specific to the diagnosis of AHF among patients in

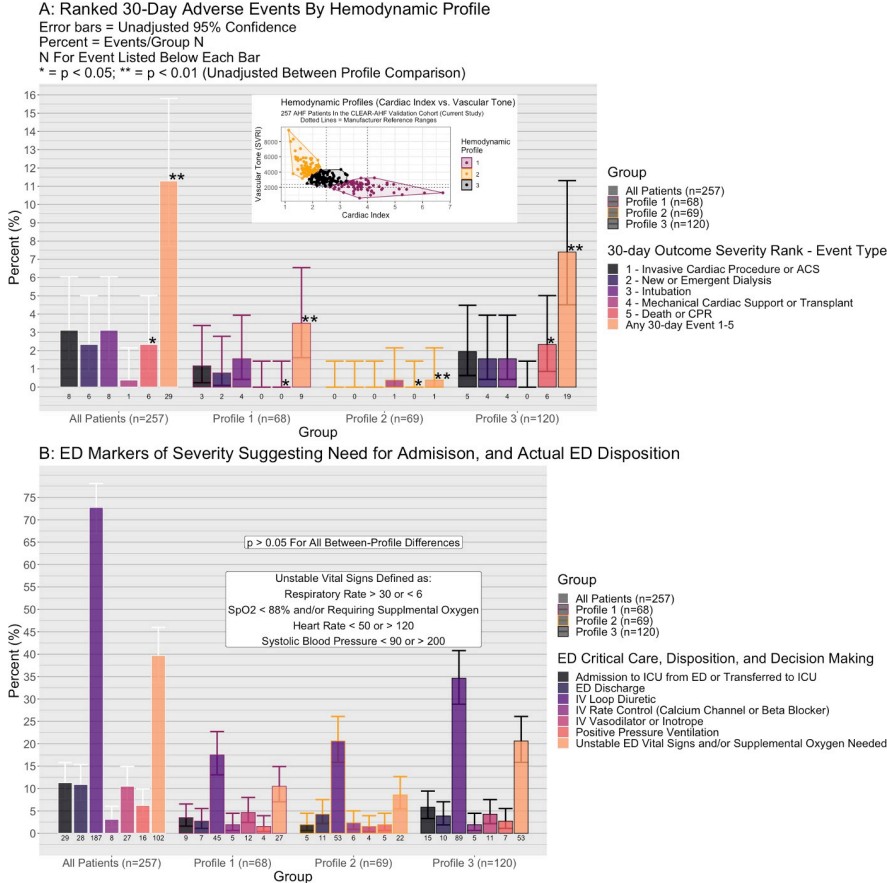

**Fig 3. Comparison of cardiac index vs. vascular tone hemodynamic profiles by 30-day adverse events, emergency department (ED) characteristics, and ED disposition.** (A) Hemodynamic profiles 1–3 in the validation cohort (inset) are compared by individual components of the composite 30-day secondary outcome. Compared to profile 2 (gold in inset), profile 3 (black) and profile 1 (purple) had greater rates of any outcome in the composite. (B) The were no statistically significant differences between profiles in actual ED disposition decisions (ICU, or discharge from ED), ED treatments administered, or the presence of unstable vital signs or need for supplemental oxygen.

the ED, as compared to one control group (CC) of septic patients and another (CC2) of patients with diagnostically-adjudicated non-cardiac dyspnea. 3. We show that these profiles have significant association with mortality and other clinically-important and patient-centered outcomes, without this association being clearly explained by other clinical variables. Overall, these results build upon the prior study [18] to suggest that these previously described hemodynamic profiles by finger-cuff have external validity, specificity to AHF, and potential clinical utility as novel markers for ED risk-stratification.

Hemodynamic profiling has long been used to subgroup AHF patients by clinically-important differences in physiology, particularly in the relationship between cardiac index and vascular tone. Despite being recommended as part of routine AHF assessment in guidelines [1,22,23], invasive measures are so specialized and uncommon [19] that ED use is virtually unheard of, while current non-invasive methods [20,21] based on physical exam are subjective and lack sufficient interrater reliability [12,18,24,41] in the ED. Finger-cuff monitors are a non-invasive approach which provides objective data, and the replication of the exploratory cluster analysis results in the prior study's DC [18] adds external validity to the observed profiles as a novel marker in ED AHF patients.

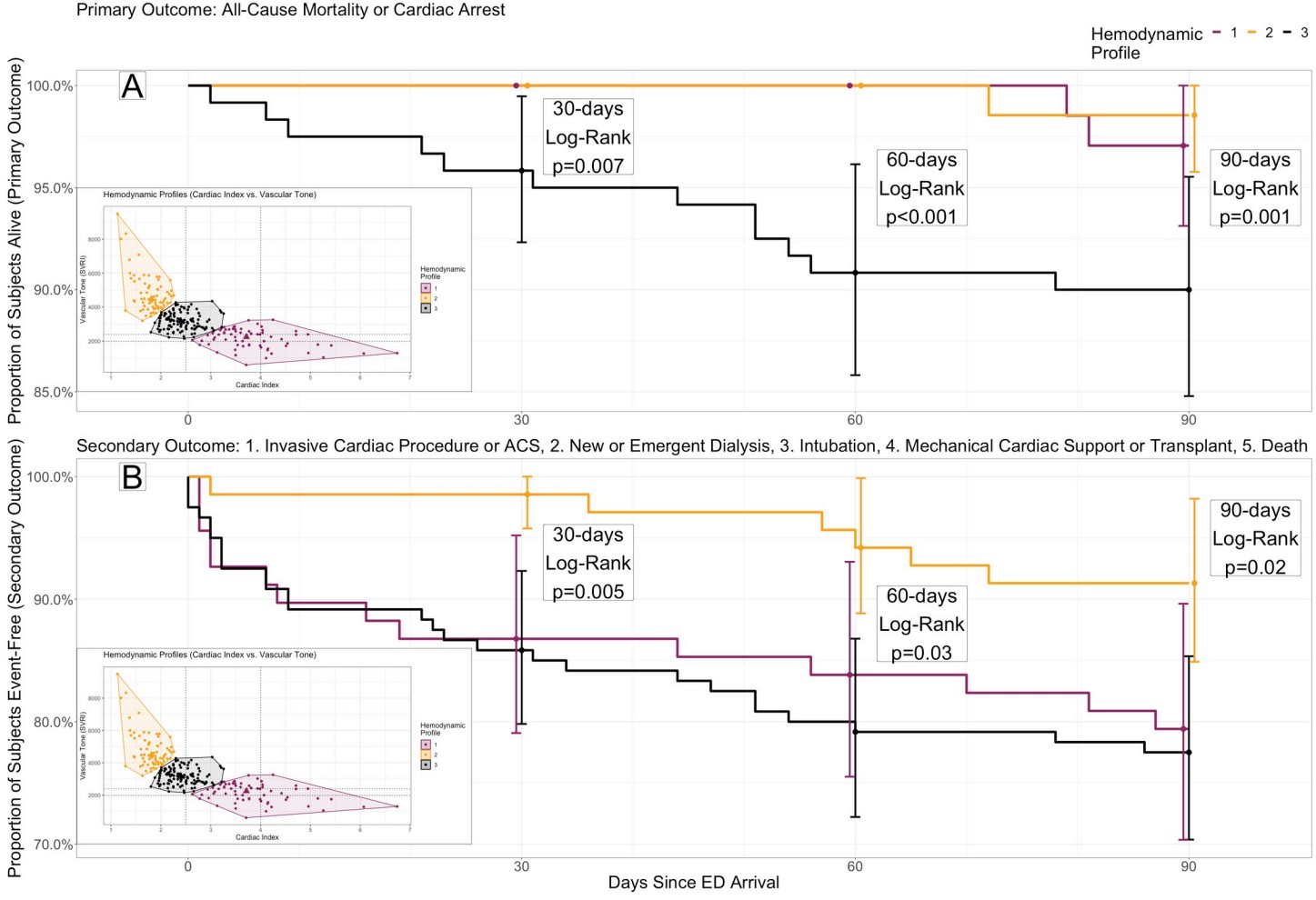

**Fig 4. Survival to primary and secondary study outcomes by hemodynamic profile.** Kaplan Meier curves for three hemodynamic profiles by cardiac index and vascular tone (see inset) in the validation cohort, through 90-day follow-up. (A) Primary outcome: All-cause mortality or cardiac arrest. Profile 3 (black) has significantly worse survival compared to profiles 1 or 2 (purple and gold, respectively) through each of 30, 60 and 90 days. (B) Secondary outcome: A composite of invasive cardiac procedure, new or emergent dialysis, intubation, mechanical cardiac support or transplant, and death or cardiac arrest. Profile 2 has significantly better event-free survival compared to profiles 1 or 3 through each of 30, 60 and 90 days.

In the prior study [18] patients had marked heterogeneity by cardiac index and vascular tone and clustered into three novel two-dimensional profiles. We were able to replicate these profiles *de novo* in a prospectively enrolled and diagnostically adjudicated external cohort, whereas the DC was a retrospective analysis without diagnostic adjudication. Additionally, we used a specialized cluster analysis procedure which was less reliant on analyst assumptions and theoretically more robust to data overfitting than what was used in the DC's profiling. Our findings nevertheless replicated the heterogeneity and distribution of cardiac index and vascular tone first noted in the DC study [18], while clearly differing from the septic CC. Replication achieved by these methods add rigor and bolster the case that these hemodynamic profiles represent a true feature of AHF patient is the ED, rather than artifact and overfitting of the prior study's single retrospective sample.

Both 90-day mortality and a composite of 30-day adverse events differed significantly between profiles in the VC. Assessing these differences was a novel and primary goal of the current study, since the DC study was neither powered for nor designed [18] to test differences

in clinical outcomes between profiles. The between-profile differences in adverse events observed were clinically significant: no patients died within the first 60 days outside the high-risk profile 3, and death remained five times more likely in profile 3 at 90 days. Additionally, the low-risk profile 2 had roughly 12 times lower likelihood of any adverse 30-day event compared to profiles 1 or 3.

Novel risk markers for AHF are needed in the ED [1], and particularly markers of low risk [1,2,8–10]. Over 80% of the 1 million AHF patients presenting to US EDs annually are admitted, including over 90% of the current sample, many of whom are at low risk for short term adverse events [1]. The burden on patients and healthcare resources is correspondingly high. Among the 141 patients in the VC who lacked one or more clear criteria for ED-to-hospital admission at our institutions (sensitivity analysis 2), the low-risk profile 2 was present in more than double the actual number of patients discharged. Similarly, over 57% of those without clear admission criteria were in profile 1 or 2, among which no patient died within 60 days. Physicians were blinded to monitoring of cardiac index and vascular tone in this study, and it is possible that knowledge of a patient's hemodynamic profile would have improved risk-stratification and facilitated ED discharges.

As in the DC study [18], hemodynamic profile was not clearly explained by other clinical variables. Two validated CDRs related to the primary outcome, and one for the secondary outcome, did not differ significantly between the high vs. low-risk profiles observed. Among a long list of common clinical variables, only BUN and history of diabetes differed between profiles. While the absence of diabetes and a lower BUN would be expected to correlate with better clinical outcomes, these variables were paradoxically the lowest in the highest risk hemodynamic profile (profile 3). Principle components analysis (sensitivity analysis 1) failed to yield any simple combination of clinical variables which would explain the variance in patients' hemodynamic profile. It is unlikely, based on the results in the VC and the prior description of the DC [18], that these hemodynamic profiles are simple functions of more common and available clinical measures. Instead, these results suggest that hemodynamic profiling by finger-cuff monitor cardiac index and vascular tone add novel information not already captured in the standard of care. Further research is needed to assess if the information added, particularly regarding association of these profiles and clinical outcomes as a risk-measure, is incremental with other established AHD risk measures. A pre-planned analysis of the VC to assess for incremental value in risk-stratification and prognosis is currently underway.

## Limitations

This study had several limitations. First, finger-cuff hemodynamic monitors have a roughly +/- 30% error in cardiac index compared to invasive monitoring [18,35], which is below the level of construct validity to completely replace invasive catheter based methods in the cardiac ICU [18,35], though relatively low as an absolute amount (mean difference in cardiac index on finger-cuff vs. invasive standard = 0.07 L/min/m$^2$ {95%CI: 0.01–0.13} [42]). The specific formula by which cardiac index, mean arterial pressure (MAP) and SVRI are calculated from the finger cuff monitor are proprietary details we are not privy to, with the only specific measures known to be incorporated being the time in systole vs. diastole, heart rate, and the velocity time integral of the pulse waveform. It is possible the device includes the calculated cardiac index and MAP in the derivation of SVRI, which would in turn extend the error rate for cardiac index to vascular tone as well. The device is highly accurate compared to invasive monitoring of MAP ($R^2$ = 0.96 [34], mean difference in MAP from finger cuff vs. invasive 4.2 mmHg {95%CI: 2.8–5.6 mmHg}), so if calculated in this way the error rate in SVRI would be unlikely to vary much from the error rate of cardiac index. Regardless, invasive monitoring is

not feasible in the ED and physical exam based non-invasive alternatives are far more unreliable and inaccurate [12,18,24,41], making the error rate in finger-cuff monitors likely the best achievable in this patient population and setting at present. Moreover, the primary goals of this study were 1. to show that clustering of ED patients by finger-cuff monitor hemodynamics were repeatable in an external sample and unique from patients without AHF and 2. to show that these profiles were associated with clinically important outcomes in ED AHF patients such as mortality and the ranked composite outcome. Both goals (and their subsequent findings) are novel compared to what has been examined in the prior literature [18] in examining extremal validity/reliability (goal 1) and predictive validity for patient-oriented outcomes (goal 2). Concurrent validity (i.e. comparison of numerical cardiac index and SVRI to a right-heart catheterization gold standard) was explicitly not a goal of this study, and may be the subject of future investigations. With this said, such a study is unlikely to be feasible or ethical in this population (ED patients with AHF and without cardiogenic shock, some of whom are discharged without hospitalization) since pulmonary artery catheterization is outside the current scope of practice of emergency medicine providers [43] and only performed in less than 1% of inpatient AHF encounters [19]. Thus, while this study was not designed to provide evidence of concurrent validity of finger cuff monitors compared to invasive monitoring, it is unlikely that such an investigation is possible in this population and this in turn underlines the potential utility of finger cuff monitoring for hemodynamic profiling of AHF. Namely, the lack of ability to employ invasive standards or useful non-invasive alternatives of hemodynamic profiling indicate a need for a novel non-invasive method feasible for the ED which is reliable between studies (i.e. external validity and reliability), consistent in differentiating profiling of AHF patients from non-AHF patients (i.e. face validity), and informative about patient-oriented and clinically-important outcomes (i.e. predictive validity) as we show here for the first time.

Second, in our approach to replication and validation using PERMANOVA means we failed to reject the null hypothesis that the VC did not replicate the DC profiles, which is not the same as accepting the hypothesis that they were the same. We used a more conservative alpha threshold of 0.3 to test this hypothesis to decrease the chance of type II error, but we cannot guarantee that additional replication studies or a larger statistical power would have failed to reject the null hypothesis. Moreover, the clear differences of the VC and CC profiles enhance our confidence in the results, given that the CC patients were enrolled at the same times and hospitals as the DC but with a different underlying ED diagnosis (sepsis, rather than AHF).

Third, our study was performed at 5 high-volume academic EDs, and results may not generalize to dissimilar settings or AHF patient populations significantly different than our sample (Table 1).

Fourth, unaccounted for lost-to-follow-up is possible for the secondary outcome, such as if a patient had an adverse event at an outside hospital. We confirmed 100% of patient follow-up for the primary outcome at 90 days between the use of telephone follow-up, HIEs including the largest hospital systems near the study sites, and dual-review with adjudication for outcome record review.

Fifth, AHF is a clinical diagnosis without a gold standard. We used double-blinded diagnostic adjudication by experienced AHF researchers and clinicians to limit the analysis of patients who could have met the inclusion criteria with signs, symptoms, and lab/imaging findings due to a diagnosis other than AHF (e.g. non-cardiac causes of dyspnea, such as chronic obstructive pulmonary disease, etc.). Moreover, a sensitivity analysis showed that profiling of patients with vs. without adjudicated AHF in the current study (i.e. included patients vs. patients excluded after adjudication as "Not AHF") was significantly different between the two groups (p<0.001). In particular, patients initially included in the current study who were adjudicated as "not-AHF" had higher cardiac index than those adjudicated as AHF (4.04 L/min vs. 3.71,

2.08 vs. 1.71, 2.88 vs. 2.4 for profiles 1–3 respectively, 2.72 vs. 2.4 overall). This adds internal validity to the diagnostic adjudication and face validity overall, by reinforcing that profiling of patients with AHF in the current study were distinguishable from those adjudicated as "not-AHF" (i.e. but otherwise meeting inclusion criteria). Diagnostic adjudication in our cohort, by standard methods for the field, is an improvement in scientific rigor over the prior study [18]. Nevertheless, the lack of a gold-standard for diagnosis of AHF (i.e. being a clinical syndrome) is a limitation to our study and all AHF literature.

Finally, while no clear combination of individual variables or CDRs appear to explain the difference in profiles, this does not imply that adding the hemodynamic profiles would add incremental prognostic value to existing AHF risk measures. Rather, incremental prognostic utility is a separate question, to be addressed in a pre-planned future analysis of the VC.

## Conclusion

In this prospective observational cohort study, we validate 3 distinct hemodynamic profiles of ED AHF patients by cardiac index and vascular tone, as measured on a non-invasive finger cuff monitor and described in prior work [18]. Mortality and a composite of adverse short-term events differed markedly between these profiles, suggesting a potential for use in the ED as a marker for risk-stratification.

## Supporting information

**S1 File. Supplemental methods.** I—Further methodological detail on consensus clustering, and contrast to k-means cluster analysis. II—Consensus Clustering Dendrograms. III—Delta-area under the cumulative distribution function (CDF) for each additional level K in consensus clustering. IV—Sensitivity Analyses—Methods, Goals, and Rationale.
(DOCX)

**S2 File. Code for analysis and minimal dataset.** Code for the R statistical programming language to reproduce results with the provided minimal datasets S3–S5 Files.
(R)

**S3 File. Minimal dataset for the validation cohort.**
(CSV)

**S4 File. Minimal dataset "VC_and_DC".**
(CSV)

**S5 File. Minimal dataset "VC_and_CC".**
(CSV)

## Author Contributions

**Conceptualization:** Nicholas Eric Harrison, Peter Pang, Phillip Levy.

**Data curation:** Sarah Meram, Morgan B. White, Sushane Gupta.

**Formal analysis:** Nicholas Eric Harrison, Xiangrui Li, Dongxiao Zhu.

**Funding acquisition:** Peter Pang, Phillip Levy.

**Investigation:** Nicholas Eric Harrison, Sarah Meram, Xiangrui Li, Morgan B. White, Sarah Henry, Sushane Gupta.

**Methodology:** Nicholas Eric Harrison, Xiangrui Li, Dongxiao Zhu.

**Project administration:** Sarah Meram, Morgan B. White, Peter Pang.

**Resources:** Phillip Levy.

**Supervision:** Dongxiao Zhu, Peter Pang, Phillip Levy.

**Validation:** Nicholas Eric Harrison.

**Visualization:** Nicholas Eric Harrison.

**Writing – original draft:** Nicholas Eric Harrison, Sarah Meram, Dongxiao Zhu, Peter Pang, Phillip Levy.

**Writing – review & editing:** Nicholas Eric Harrison, Xiangrui Li, Morgan B. White, Sarah Henry, Sushane Gupta, Peter Pang, Phillip Levy.

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
