## [Decision Letter · Decision Letter 0]

7 Jan 2022

PONE-D-21-36949Hemodynamic profiles by non-invasive monitoring of cardiac index and vascular tone in acute heart failure patients in the emergency department: external validation and clinical outcomesPLOS ONE

Dear Dr. Harrison,

Thank you for submitting your manuscript to PLOS ONE. After careful consideration, we feel that it has merit but does not fully meet PLOS ONE’s publication criteria as it currently stands. Therefore, we invite you to submit a revised version of the manuscript that addresses the points raised during the review process.

We look forward to receiving your revised manuscript.

Kind regards,

Gianluigi Savarese

Academic Editor

PLOS ONE

Journal Requirements:

Reviewers' comments:

Reviewer's Responses to Questions

**Comments to the Author**

1. Is the manuscript technically sound, and do the data support the conclusions?

Reviewer #1: Yes

Reviewer #2: Partly

2. Has the statistical analysis been performed appropriately and rigorously? 

Reviewer #1: Yes

Reviewer #2: Yes

3. Have the authors made all data underlying the findings in their manuscript fully available?

Reviewer #1: Yes

Reviewer #2: No

4. Is the manuscript presented in an intelligible fashion and written in standard English?

Reviewer #1: Yes

Reviewer #2: Yes

5. Review Comments to the Author

Reviewer #1: This manuscript by Professor Levy et al. addresses the use of a non-invasive finger-cuff monitor measuring CI and SVRI in AHF patients in an ED setting. The research group is well established in the field. The manuscript is well written, and a lot of effort has gone into the work.

In summary, my main criticism of this work relates to the limitations of finger-cuff monitors. 1) The authors do discuss this as a limitation, with a roughly +/- 30% error in CI compared to invasive monitoring. However, the calculations of SVRI are also based on CI, resulting in a higher risk of error, which is not addressed. I also wonder if the results to a large extent are associated with blood pressure. What is the clinically additional value of CI and SVRI? Figure 2 shows correlations which are well known: The higher the SVRI (afterload), the lower the CI (CO).

2) Secondly, I wonder why the authors have chosen prespecified clusters instead of clustering all patients de novo. In my opinion, this would have been preferable. I understand the rationale behind prespecified clusters but suggest a reanalysis.

3) Thirdly, in table 1, I don´t see that multiple hypothesis tests have been performed. Were there any significant differences?

4) Lastly, although validation of previous published data could be valuable, the additional new information from this study is limited.

Reviewer #2: The authors presented a validation study of a novel non-invasive monitoring technique for the hemodynamic profiling of patients with AHF. The study has a good rationale and novel techniques are strongly warranted in the management of patients with AHF. However, I have several concerns related to the structure of the study:

- although this is a validation study, the definition of hemodynamic profile in AHF should require, at least for a subgroup of patients, direct invasive hemodynamic data for comparison

- the control cohort of patients with sepsis might not be optimal and an additional comparison with healthy individuals would be appreciated

- the diagnosis of AHF is clinical, the authors should reinforce (or acknwoledge in limitation) the criteria for inclusion as the risk of including patients with non-cardiac dyspnea is not minimal

- the text is too long, in particular the introduction and methodological section, and sohuld be shortened and made clearer for clinical readers

6. PLOS authors have the option to publish the peer review history of their article (what does this mean?). If published, this will include your full peer review and any attached files.

Reviewer #1: No

Reviewer #2: No

---

## [Author Response · Author response to Decision Letter 0]

8 Feb 2022

January 9, 2022

RE: PONE-D-21-36949, “Hemodynamic profiles by non-invasive monitoring of cardiac index and vascular tone in acute heart failure patients in the emergency department: external validation and clinical outcomes”

Dear Dr. Savarese, our peer reviewers, and PLOS ONE staff,

On behalf of myself and my co-authors, thank you for your time and effort in considering our manuscript the opportunity to revise it for publication. In our "Response to Reviewers" document included in the resubmission, we have offered our responses to each of the reviewer comments including line numbers referencing the edits made in the manuscript. We have further included the revised manuscript in two versions, with and without changes tracked, in the resubmission. 

Sincerely,

Nicholas Eric Harrison, MD, MSc

Assistant Professor of Emergency Medicine

Indiana University School of Medicine

REVIEWERS' COMMENTS:

Reviewer's Responses to Questions

Comments to the Author

1. Is the manuscript technically sound, and do the data support the conclusions?

Reviewer #1: Yes

Reviewer #2: Partly

- Thank you for this feedback. Please see our responses to your comments below and the corresponding edits, and please let us know if there are any further concerns.

2. Has the statistical analysis been performed appropriately and rigorously?

Reviewer #1: Yes

Reviewer #2: Yes

3. Have the authors made all data underlying the findings in their manuscript fully available?

Reviewer #1: Yes

Reviewer #2: No

- Thank you for this feedback. The full dataset and the R code to reproduce the statistical analyses were included in the original submission. Please let us know if there is something missing from the uploaded dataset and the R code we may have missed, so that we may provide anything further that is needed. Our hope has been to ensure we meet the highest standards of data transparency possible for this publication.

4. Is the manuscript presented in an intelligible fashion and written in standard English?

Reviewer #1: Yes

Reviewer #2: Yes

5. Review Comments to the Author

Reviewer #1: This manuscript by Professor Levy et al. addresses the use of a non-invasive finger-cuff monitor measuring CI and SVRI in AHF patients in an ED setting. The research group is well established in the field. The manuscript is well written, and a lot of effort has gone into the work. In summary, my main criticism of this work relates to the limitations of finger-cuff monitors. 

1) The authors do discuss this as a limitation, with a roughly +/- 30% error in CI compared to invasive monitoring. However, the calculations of SVRI are also based on CI, resulting in a higher risk of error, which is not addressed. 

Thank you for this feedback. We agree this is also a limitation, and like most of the comments here, much was left out of this paper to that effect because it was already quite long (as reviewer 2 points out). However, we understand the concerns raised and have tried to add back an extensive amount to the limitations section to address this. In response to reviewer 2’s feedback, we have moved a great deal to the supplement, while also adding text to meet the requested changes by yourself and reviewer 2. We have tried our best overall to strike a balance between parsimony and being thorough, so please let us know (for the edits in response to this comment as well as all the others) whether more detail in the text is required because we can always add more. In many cases we left language out we would have liked to include simply in an effort to meet the request for more brevity.

With regards to the specific point raised here - we actually cannot be completely certain if this is or is not how the machine calculates SVRI (i.e. by first calculating CO and MAP, and then deriving it by SVR = MAP/CO). The manufacturer keeps this information proprietary, and we are unfortunately not privy to the specific details. The details that are public include that CO is calculated from the systolic pressure time integral of the arterial pulse waveform and heart rate, plus a proprietary model adjusting for physiologic variables including age, gender, and of course height and weight. Please see https://education.edwards.com/clearsighttm-system-technology-overview/258045#. We have added text at line 601-611 to clarify that the specifics of this formula are unknown besides the basics (i.e. that the volume clamp method is used to construct a continuous arterial pulse waveform, integrated to derive a stroke volume, and adjusted by a proprietary formula).

However, to your point, we can at least speculate that the device likely calculates CO and MAP first, and thereafter calculates SVR after this from these two values (and SVRI by including the body surface area adjustment). We have added language from lines 601-680 to help clarify to the reader what the implications would be if this is indeed a correct assumption on how SVRI is calculated, and what would be expected as far as error in SVRI. Overall, the contribution of error from MAP is likely nominal because the device has R2 = 0.96 agreement (interclass correlation) in blood pressure compared to invasive monitoring (i.e. arterial catheter) as referenced in the added text. A metaanalysis (Saugel et al 2020, referenced in the added text) found that the absolute mean difference between the finger-cuff device and invasive monitoring for MAP was +4.2 mmHg (95%CI: 2.8-5.6 mmHg). Thus, since this error is quite small in a clinical sense, the error in SVR (and therefore SVRI) likely comes predominantly from the known error in CO (and is thus likely to be of a similar magnitude, i.e. +/- 30%). 

We have also added text at line 601-602 to clarify the error of the CO in absolute terms. While the percent error is around ~30% as previously referenced in the text, the absolute value of error between finger-cuff and invasive cardiac index is about 0.07 L/min/m2 {95%CI: 0.01-0.13} (Saugel 2020 metaanalysis, added as a reference in lines 601-602). From the standpoint of clinical significance, and in comparison to the overall spread of measured cardiac index in the sample, this is quite small. For instance, the IQR of cardiac index in the sample and for each profile were as below - 

All Patients {n=257} 2.4 L/min/m2 ; IQR: 2.00-3.08

Hemodynamic Profile 1 {n=68} 3.71 L/min/m2 ; IQR: 3.35-4.12 

Hemodynamic Profile 2 {n=69} 1.81 L/min/m2 ; IQR: 1.64-1.97

Hemodynamic Profile 3 {n=120} 2.4 L/min/m2 ; IQR: 2.18-2.65

On an absolute basis, an error rate of 0.07 L/min/m2 compared to invasive monitoring, as cited in the added text, is relatively small compared to the spread of data overall and the differences between each profile.

Finally, one of the reasons we left out some of this context from the limitations section in the original submission (in addition to a long text, as reviewer 2 points out) is that ultimately, we were explicitly not trying in this study to validate the finger cuff monitor for SVRI or cardiac index measurement (i.e. as compared to pulmonary artery catheterization). Please see responses to reviewer 2 comments about comparison to invasive monitoring for more detail here. In brief - we have added text to be more explicit that: 1. A comparison to an invasive standard was not a goal or a finding of this study 2. While such a comparison could be a goal of future study, it is highly unlikely such a study could be approved in this population (ED patients before disposition, including those who may be discharged without admission) since invasive SVRI and cardiac index monitoring (with a pulmonary artery catheter) is outside the scope of practice for emergency physicians and performed less than 1% of the time even among AHF patients after admission, 3. The goals of this study were to show that A. Hemodynamic profiling by the finger cuff device could externally validate the prior study (Nowak 2017 as cited in text), B. This profiling, regardless of its error rate in SVRI and cardiac index compared to invasive monitoring, nevertheless predicts patient-oriented outcomes which could be useful to emergency physicians as a novel tool for risk stratification (see further discussion and edits regarding your question about novelty and impact). 

2) I also wonder if the results to a large extent are associated with blood pressure. 

Thank you for this comment. This is certainly a possibility to consider which we had a lack of clarity on as currently written, and have added further language at lines 467-472 to add clarity. As mentioned, there was no significant difference between profiles (see table 1) for blood pressure.

Overall, differences in SBP and DBP were neither statistically significant nor relatively different in a clinically-significant manor. DBP was virtually identical across profiles. The profile (1) with the highest SBP had the lowest SVRI, and the profile (2) with the next highest SBP had the highest SVRI. At the same lines we have added clarification from a recent reference, where finger cuff monitoring was used in the same population, and a Bayesian multivariate analysis showed that the finger cuff SVRI was only weakly correlated with MAP (r = 0.479, 95% interval 0.379-0.568) and SBP (r = 0.324, 95% interval 0.206-0.426). SVRI on the finger cuff was moderately correlated with DBP (r = 0.587, 0.498-0.660), but again the profiling had virtually no discernable difference by DBP (let alone a statistically significant one).

 For background, the point of the above prior analysis whose reference we have added at the lines indicated, was because we wondered exactly the same questions as you have presented. Namely, if SVRI (or cardiac index) were strongly associated with blood pressure in a highly predictable way it would lessen the utility of the finger-cuff monitor. As we found out, SBP was just as strongly correlated with on the finger cuff monitor in this population with cardiac index (r=0.295, 0.181-0.404) as it was SVRI. 

3) What is the clinically additional value of CI and SVRI? Figure 2 shows correlations which are well known: The higher the SVRI (afterload), the lower the CI (CO).

Thank you for pointing out that this was unclear. Similar to comments elsewhere, we have added text to better clarify what was and was not a goal of the study. Showing the simple relationship between cardiac index and SVRI was not a goal of this study - this has already been done on the finger-cuff monitors in this population in the original Nowak et. al 2017 study (Am J Emerg Med. 2017;35(4):536-542) and as you correctly point out is simply an expected physiologic relationship regardless (i.e. MAP= CO x SVR). Figure 2 was included to give graphical/visual comparison for the external validation (and replication) aim of the study - i.e. showing in panel 2B that the clusters of observations look similar between the VC and the DC, but the clusters of observations of the VC and non-AHF patients in the CC look different. In other words, showing that in this study we validated the clustering (i.e. 2-dimensional/multivariate distribution on cardiac index and SVRI) of AHF patients in a new/external population compared to the original Nowak study, this in turn shows that the clusters first described/derived in Nowak 2017 are able to be replicated. The rationale here is this: if the finger-cuff is to at all have any clinical utility (in the ED, in AHF patients) the profiling must be able to be reproduced (externally-validated) between samples. An inability to replicate profiling among AHF patients in a new sample would suggest that the profiling in Nowak 2017 was simply random statistical noise (i.e. rather than a generalizable feature of AHF patients in the ED compared to non-AHF patients, see response to Reviewer 2 inquiry regarding comparison to non-cardiac dyspnea), and therefore any clinical use of the finger cuff monitor would be effectively moot. An important point to understand here (and which hopefully we have helped clarify with the edits to the text) is that again we assume nothing here about the accuracy compared to an invasive monitor - i.e. concurrent validity - beyond what is reported in the prior literature (all outside the ED setting, for reasons discussed in the response to reviewer 2). Rather, we only were interested in between-study reliability of the clustering by the monitor (i.e. external validity). Showing reliability/external validity makes it plausible that the finger-cuff monitor could be used in the ED to obtain numbers that are similar regardless of site/center/local population of AHF patients, and adds rigor to the very exploratory and descriptive results shown in the prior study (Nowak 2017). See later answers to your question about novelty/clinical impact for more detail on this. 

To your original question, given the above - Figure 2 is therefore simply an effort to give the reader a sense of how closely things were able to be replicated in the VC compared to the DC (Nowak 2017 AHF patient original clustering), and the CC (septic controls enrolled at the same time and from the same places as the Nowak 2017 AHF patients). Put simply: in an external sample, AHF patients had similar clustering on the finger-cuff as the prior study of AHF patients (Figure 2B), and clustering was visibly different from patients with sepsis (which would be expected since sepsis has very different hemodynamics than AHF, figure 2C). Note in response to a comment from reviewer 2 we also performed an additional analysis comparing AHF patients in the VC to those patients enrolled at the same time and adjudicated to have non-cardiac dyspnea (see response to Reviewer 2 below), to give an additional control. 

The visual comparison in figure 2 is simply qualitative, and of course we showed (as described in the text) quantitatively using PERMANOVA that clustering in the VC was similar to the DC and different in a statistically-significant way from the CC (as well as the non-cardiac dyspnea cohort we added in response to reviewer 2). The point of figure 2 beyond the quantitative analysis reported in the text was twofold: 1. We assumed many readers would find a visual comparison helpful, since it is hard to understand intuitively the multivariate (i.e. a 2-dimensional response variable) statistical comparison we made to perform the validation 2. For someone reading the paper to be able to read an output from the finger-cuff monitor and directly pinpoint which profile their patient was in based on the cardiac index and SVRI numbers (by just looking where in figure 2A their patient’s numbers corresponded to). For similar reasons as #1 (difficulty of simply “thinking through” a reference range in multi(2)-dimensional space), it would be less difficult for a busy clinician to figure out which profile their patient fits into without a visual aid.

2) Secondly, I wonder why the authors have chosen prespecified clusters instead of clustering all patients de novo. In my opinion, this would have been preferable. I understand the rationale behind prespecified clusters but suggest a reanalysis.

Thank you for this comment. All patients were, in fact, clustered de novo. We have edited for clarity at line 305. The supplement contains the detailed results of the de novo clustering (placed here because it was felt the technical detail of consensus clustering was a bit much for the typical reader within an already quite long paper). Please let us know if further clarity is needed, if more detail is needed, and/or if a specific reanalysis of the (de novo) consensus clustering we performed is still needed following this clarification (i.e. something perhaps which we have not considered or misunderstood from the comment here). We appreciate you pointing out that this was unclear before.

4) Thirdly, in table 1, I don´t see that multiple hypothesis tests have been performed. Were there any significant differences?

Thank you for this comment. We certainly could apply an adjustment of significance for multiple comparisons to table 1. However, we should note that all standard statistical methods which adjust for multiple comparisons do so with the result (and the goal of) reducing the significance of differences (i.e. all result in it being less likely to reject the null hypothesis, rather than more, to limit spurious discovery of a significant difference). Please let us know if there is a specific technique you mean and would like us to apply (i.e. perhaps one of which we are unaware) which adjust for multiple comparisons and result in the non-significant differences throughout table 1 potentially becoming significant. 

5) Lastly, although validation of previous published data could be valuable, the additional new information from this study is limited.

Thank you for this feedback. We have added an extensive amount of text from lines 156-174, 526-597 and 616-640 and 657-680 to clarify the goals of the study and why the results from these goals are relevant. To be honest, however, there is far more detail yet we could go into towards this end (as evidenced by the unusual length of this response letter, for which we apologize) and once again should be clear that we wanted to balance addressing these suggestions with the desire for more brevity in the paper from reviewer 2. Please let us know if more detail is needed in considering our response as we are happy to add more and/or revise further. 

To your specific point about clinical impact - 

First, it certainly is true that even if the finger cuff monitor produces reliable/consistent clustering in separate studies in an ED AHF population (i.e. external validation of the profiling with this machine in this population, our first objective), this says nothing about the clinical utility of this tool. Namely, a reliable tool (even one reliable across studies/populations) is only useful if it also has validity for some clinically useful goal. In other words, even if profiling as described first by Nowak et al is externally validated as we showed, this is only clinically impactful if that reliable information can be used towards a clinically useful goal. In light of this, we could have defined a “clinically useful goal” in either of two ways: 1. concurrent validity (i.e. the finger-cuff monitor has strong accuracy compared to invasive monitoring on a pulmonary artery catheter), or 2. the profiling as described first in Nowak 2017 and externally validated here is associated with patient-oriented outcomes (i.e. predictive validity). We explicitly decided that #1 (concurrent validity) could not be a goal of this study due to the reasons discussed in the response to Reviewer 2’s comment about comparison to an invasive standard - e.g. invasive hemodynamic monitoring is outside the scope of practice of emergency physicians and therefore never performed in AHF patients prior to ED disposition, and it is rarely performed even after hospital admission (~1% of contemporary AHF admissions {Hernandez 2019 Journal of cardiac failure. 2019;25(5):364-371}) and generally reserved for the sickest or most clinically complex patients. The latter introduces spectrum bias, compared to the other 99% of admitted AHF patients, and certainly in comparison to those never admitted (i.e. those treated and evaluated in the ED only). Put simply, it would be nice to show that cardiac index and SVRI correlate strongly with a pulmonary artery catheter, but such a study is likely infeasible and unethical in ED patients before disposition (since emergency physicians do not have invasive hemodynamic monitoring in their scope of practice, and invasive monitoring with a pulmonary artery catheter does have greater than minimal risks). Moreover, in our view, the large amount of literature out there comparing finger-cuff monitors to invasive monitoring (all in settings where the clinicians do have pulmonary artery catheterization in their scope of practice, namely the OR and ICU) would actually make a comparison to invasive monitoring in an ED setting less novel and of less clinical benefit, since invasive monitoring isn’t even an option in the ED to begin with, and accuracy compared to invasive SVRI or cardiac index is certainly not a patient-centered outcome. All of which brings us to goal 2 - i.e. do these profiles on the finger cuff monitor, first described in the ED population of AHF patients by Nowak et. al and able to be externally validated here, have predictive validity for patient-oriented clinically important outcomes? As we describe in our introduction and elsewhere (but which was pared back since, as reviewer 2 points out, the paper is already quite long) risk stratification of AHF patients by emergency physicians before disposition is one of the greatest unmet challenges in ED care of AHF. The HFSA and SAEM guidelines on early management of AHF (Collins 2015, as referenced in manuscript) singled out the identification of novel tools for ED risk stratification as a major unmet need. Finger-cuff monitoring in the ED, and particularly using the profiling first described in Nowak 2017 and validated here, is certainly a novel method: finger cuff monitors are used in ED patients prior to disposition as often as pulmonary artery catheterization, which is to say never. Therefore, if this novel method was reliable and externally valid (our objective 1) and it corresponded to clinically important outcomes (like mortality and the other adverse events we examined) this could in turn aid ED risk-stratification (predictive validity, our objective 2). We would argue that this would be a clinically impactful result regardless of how closely it could be compared to an invasive standard which cannot be used in the ED anyway (i.e. concurrent validity, which was not a goal of the study). Certainly, it would be more patient-oriented, at least.

Second, note that once again because of the already great length of this paper and the number/complexity of the statistical analyses already therein, a separate analysis which further highlights the clinical impact is to be published separately. We allude to this in the limitations section: even though the profiling was associated with clinically important outcomes and no other clinical variables clearly distinguished the profiles, this only suggests in a basic way that the prognostic (risk-stratification) value of the profiling is new information when added to standard clinical variables obtained in the ED. Showing incremental predictive value - i.e. a formal multivariable hypothesis test to see if this profiling adds predictive value for mortality and other adverse events after adjusting for the predictive value of current clinical standards in ED-based AHF risk-stratification - is a separate aim involving several additional analyses which we performed but nevertheless felt were too much to include in 1 paper (pursuant to Reviewer 2’s comment that the paper is already too long without any of that). In that analysis (planned to be published separately) we found that the prediction for mortality and other adverse events by profile remained significant after adjusting for multiple validated AHF clinical risk scores (e.g. Get With the Guidelines Heart Failure risk score, Emergency Heart Failure Mortality Grade, STRATIFY by Collins et al.). The profiling appropriately reclassified a significant number of low-risk patients who were erroneously classified as high-risk by these risk scores/clinical decision instruments. In other words, several patients whom the validated decision instruments would have classified as high risk prior to ED disposition (and therefore whom the ED clinician would certainly would have admitted to the hospital), but who in actuality had no adverse events, would have been reclassified as low-risk by use of the finger-cuff profiling. In turn, these patients would have been able to have been identified as appropriate for observation unit or even possibly direct discharge from the ED thanks to the added information of the hemodynamic profiles we evaluated.

To your point about the clinical impact of the current paper, we do think this separate analysis regarding incremental prognostic value (as briefly described above) is the more clinically-impactful finding. However, as mentioned, putting all of what we report in the current paper in a single manuscript with this additional analysis was felt simply to be too much (both for size of the manuscript, and ease of interpretability). We also considered dropping from the manuscript the reporting on externally validation of the Nowak 2017 clustering (objective 1), and/or the unadjusted associations with clinical outcomes between profiles (objective 2) which make up the current manuscript in favor of just reporting the analysis of incremental prognostic utility (“objective 3” as described above, and planned as of now as a separate manuscript). However, we felt the current approach to be better for a few reasons: 1. Incremental prognostic value and clinical utility for risk stratification (objective 3, forthcoming manuscript) means nothing if the original profiling done by Nowak in 2017 turned out to be spurious (i.e. without external validity, unable to be replicated) 2. Similarly, we felt it was important to compare the AHF profiling to a control cohort (the septic CC in the original draft, and in response to reviewer 2 the additional control cohort of patients with non-cardiac dyspnea). Incremental prognostic utility of the profiling (and the profiling overall) would have less face validity if it was not able to be shown that the profiling of AHF patients was different from non-AHF patients 3. The current manuscript directly responds to/addresses what Nowak et. al identified as the primary limitations of their original study, and thereby is a natural next step in evaluating this paradigm in a way that improves upon the scientific rigor of what came before. Because of each of these reasons, we felt publishing the results in the current manuscript a necessary step to establish the rigor and validity underlying the more clinically relevant analysis about incremental value in risk-stratification.

To the latter point, prior to the original study [Nowak et. al, 2017, Am J Emerg Med. 2017;35(4):536-542] quantitative cardiac index and SVRI monitoring had not been described previously in an ED population of AHF patients, and while the results were interesting as a descriptive/exploratory analysis, there were legitimate methodological concerns which the authors identified as needing addressed in future study (i.e. this one) to strengthen the methodological rigor and certainty of inference. First, Nowak et al identified that their analysis was limited by not having any diagnostic adjudication. AHF is a clinical diagnosis and uncertainty in diagnosis (and therefore study inclusion) is one of the greatest limitations of all AHF literature. As reviewer 2 points out, even our current diagnostically-adjudicated study (e.g. versus non-cardiac dyspnea) has this limitation, and it is certainly true that the completely unadjudicated cohort in Nowak 2017 was even more limited. Thus our work here is an intentional improvement in methodological rigor over the prior study. Second, Nowak et. al noted that their study was underpowered to detect any between-profile differences in patient-oriented clinically-significant outcomes. Because of this, our study was designed to be able to detect such differences (i.e. objective 2), with more rigorous methods of outcome collection in addition to a larger sample size/more statistical power, as an explicit response to the Nowak 2017 study. 

With all that said, we felt publication of the current results (i.e. because of how they establish rigor for the more clinically-important analysis, by improving upon the primary limitations in the previous study) was necessary. Moreover, due to this being reported separately from the more clinically-focused analysis, we felt this manuscript appropriate for a journal like PLOS ONE where clinical impact is not among the criteria considered for publication. Rather, it is our understanding of the journal’s mission and publication criteria that the focus is on rigor of the underlying methods and interpretation of results, without regard to clinical impact. However, if it is felt necessary and not too burdensome to an already long manuscript, we can certainly add the further analysis regarding incremental value in risk-stratification in to what is here currently. Please let us know if this is desired. 

Thank you for your time in peer reviewing and your thoughtful comments.

Reviewer #2: The authors presented a validation study of a novel non-invasive monitoring technique for the hemodynamic profiling of patients with AHF. The study has a good rationale and novel techniques are strongly warranted in the management of patients with AHF. However, I have several concerns related to the structure of the study:

- although this is a validation study, the definition of hemodynamic profile in AHF should require, at least for a subgroup of patients, direct invasive hemodynamic data for comparison

Thank you for the feedback. We thought a lot about this too, much of which was ultimately removed from the manuscript because (as you point out later) this manuscript is already quite long. We have added back a large amount of text from lines 601 -680 discussing this point in the limitations section, and to hopefully reinforce a lack of comparison to an invasive standard as a limitation of the study.

From line 601-680 we have added text to help clarify that finger-cuff monitors have been compared with an invasive gold standard in numerous past studies for cardiac output/cardiac index (i.e. pulmonary artery catherization) and blood pressure (i.e. invasive blood pressure monitoring) in the ICU and perioperative/OR setting . We added a reference to a 2020 metaanalysis (Saugel 2020) summarizing these studies comparing finger cuff monitors to a gold standard. We added a clarification on the error rate for cardiac index on the finger-cuff monitor (based on the several studies in the metaanalysis with comparison to invasive methods) in terms of the absolute difference between the finger-cuff and pulmonary artery catheter thermodilution (+0.07 L/min/m2, 95%CI: 0.01-0.13) which likely better gives context to the degree of error than percent difference. Additionally, we clarify in the added text the accuracy of the finger cuff for MAP, which as previously cited in our introduction represents an extremely high intraclass correlation for invasive vs. non-invasive (R2=0.96). Please see the comments in response to reviewer 1 for more discussion on the above. 

Despite the numerous prior studies comparing the finger-cuff in the ICU and OR settings as outlined in the newly cited metaanalysis and clarified text, we agree that in an ideal world a comparison to pulmonary artery catheter monitoring would be helpful, since despite the aforementioned evidence there are no studies known to us comparing finger-cuff monitors to invasive monitoring in the ED setting specifically. As mentioned above, we considered this, but ultimately a comparison compared to an invasive standard was deemed infeasible and likely unethical in the ED setting, since pulmonary artery catheterization is not performed in the emergency department. We have added a citation in these lines to help clarify that pulmonary artery catheterization is not within the scope of practice of emergency physicians. Further we have added text in the same lines to clarify that as rarely as invasive hemodynamic monitoring is performed in inpatients (1% of contemporary AHF hospitalizations, as referenced in the text), it is never performed in the ED among patients prior to disposition (i.e. those who are not yet admitted to the hospital and may in fact even be discharged without hospitalization, our study’s population) because of the lack of training and expertise among emergency physicians to perform this procedure and the risks of the procedure even when performed by specialists who do receive training to perform it (e.g. cardiologists, intensivists, surgeons, and some anesthesiologists). In the lines mentioned for the added text, we briefly clarify the goals of the study and the reason why, explicitly, a comparison to an invasive standard was not a goal and how this exists as limitation. 

In short, recognizing that a comparison to invasive monitoring (i.e. concurrent validity) in the ED prior to disposition was unlikely feasible or ethical, we sought in this study to test whether the finger-cuff hemodynamic profiles provided predictive validity (i.e. ability to predict clinically relevant and patient-centered outcomes like mortality). As discussed in the responses to reviewer 1, a lack of statistical power and methodological rigor to test differences in clinical outcomes between the finger-cuff profiles was named as a primary limitation of the original study (Nowak 2017, as cited in text). The naming of this as a limitation and unanswered question of the Nowak 2017 study was one of two primary justifications for the current study - i.e. the current study was designed explicitly to be able to better assess for clinical outcome differences between profiles as we showed. This goes along with the other primary goal of this study (to externally validate the finger-cuff based hemodynamic profiling first described in Nowak 2017) for the overall goal of determining whether the profiles first described in Nowak 2017 have clinical utility for risk-stratification of AHF patients prior to ED disposition decision to admit or discharge the patient (i.e. one of the greatest current needs in ED evaluation and management of AHF, as cited/described in our introduction, though only briefly because as you mention later the introduction is already quite long). Please see responses to reviewer 1 for more details.

Ultimately our rationale for seeking to test predictive validity (do profiles predict clinically-important patient outcomes?) and not concurrent validity (does the finger cuff monitor produce the same SVRI and cardiac index measurements in ED patients as a pulmonary artery catheter) is perhaps best explained first through an analogy: An ECG obviously is not as accurate at predicting the presence or absence of acute myocardial infarction (AMI) in ED patients with chest pain as it would be for the physician to perform a cardiac catheterization in all those presenting to the ED with chest pain. However, there is no such thing as an emergency physician performed cardiac catheterization (let alone one performed at the point-of-care/bedside, because it would be unsafe and outside the scope of practice of emergency physicians just like pulmonary artery catheterization). For this reason, no study has ever compared ECG versus the gold standard of cardiac catheterization at the bedside, and in the ED. Even if a study comparing ED-performed cardiac catheterization to ECG were feasible by limiting the comparison to only the highest risk population (e.g. only those with a dynamic troponin elevation, wall motion abnormalities on echocardiography, etc.), it is questionable how useful this would be since restricting the comparison would result in spectrum bias - i.e. the patient population would fairly closely represent patients admitted to the hospital or ICU with an already high suspicion (pretest probability) of AMI. Since comparison of ED ECG to subsequent performance of left heart catheterization after hospital admission and with a high pretest probability has already been extensively studied, this hypothetical ED-based study involving only the highest-risk patients would provide little new information to what is known. Furthermore, we know quite clearly that ECG abnormalities in the ED do not perfectly predict intervention-amenable coronary occlusion being detected on those patients who the emergency physician decides to admit to the hospital, and whom the cardiologist ultimately decides to take for cardiac catheterization. Many ECG findings which can relate to AMI are non-specific and/or insensitive. Yet despite this known lack of accuracy and no past studies directly comparing to a gold standard in the ED setting for a broad population of ED chest pain patients across the risk spectrum, the ECG is nevertheless still a critical tool in the initial ED evaluation and risk-stratification (including decision to admit patient to hospital, consult a cardiologist, get further testing, or simply discharge directly from the emergency department). This is because it does have some predictive value for AMI on later cardiac catheterization, it is able to be obtained by a nurse at the bedside quickly and non-invasively so as to be feasible in all patients presenting to the ED with chest pain, and obtaining the gold standard of cardiac catheterization in an ED patient is not feasible for emergency physicians to use. Nor would it be ethical to test cardiac catheterization in comparison to a screening test like ECG in this population (i.e. a broad swath of ED patients with chest pain across spectrums of risk) since the vast majority of ED chest pain patients do not have AMI and do not need invasive management or even hospital admission. 

This is analogous to the issue with the finger cuff monitor: 1. We know it is not perfectly accurate for cardiac index and SVRI (error rate of +0.07 L/min/m2 {95%CI: 0.01-0.13} for cardiac index, Saugel 2020 as referenced in edited copy) in the ICU and OR setting, where the gold standard is within the clinical scope of practice (e.g. by the intensivist or cardiologist or other specialist who receives training in this procedure, unlike the emergency physician) 2. We know that the gold standard (again cardiac catheterization, though in this case a right-heart cath) is not in the scope of practice of ED clinicians (save those few with concomitant ICU fellowship training) and likely not feasible or ethical as a screening procedure in all AHF patients regardless (since the literature would suggest that as many as half of the 1 million AHF patients who present to US EDs annually have preventable admissions and low-risk phenotypes, and who potentially should be discharged rather than admitted) 3. Our results suggest that the profiling first described by Nowak et al in AHF patients in the ED is repeatable and consistent in an external population from the original derivation 4. Our results suggest that these profiles are significantly associated with relevant clinical outcomes like mortality. Taking point 1 and 2 together, it is unlikely a study comparing invasive monitoring to the finger-cuff in an ED population before disposition could currently meet the threshold of clinical equipoise needed to make such a study ethical, especially given an already large body of literature in other settings like the ICU and OR. On the other hand, taking points 3 and 4 together, whether the invasive to non-invasive correlation in cardiac index is different in the ED compared to the ICU or OR is effectively a moot point, since invasive options are not even possible in this population and the non-invasive option does appear to be reliable and have clinical utility for patient-oriented outcomes. Put another way, the goal of the study was not to show that the finger-cuff could be used to detect clinically-significant profiles that were the same as profiling on an invasive monitor, but rather just to show that the profiles the finger-cuff does detect may have some clinical utility (i.e. by predicting mortality and other outcomes) and be reliable (i.e. through external validation). 

We apologize if this was not entirely clear and hope some of what we have added in the edits have helped clarify this, and highlight both the limitations and strengths of the study in an appropriate way. However, we are also highly cognizant of how long and methodologically complex the manuscript already is. As mentioned in the comments to reviewer 1, we originally intended to include even more detailed analyses on the prognostic value of the finger-cuff profiles in this manuscript but (as you point out later) the manuscript already is long enough in the methods and introduction to be somewhat difficult to read without add more.

To the question of performing a comparison to an invasive standard in only a select few patients - It is theoretically possible that a future study could overcome the ethical and feasibility barriers of an ED-performed comparison of the finger cuff to right-heart catheterization by limiting it to only the sickest/highest risk AHF patients. However, like the ECG example, this would in effect defeat the purpose: restricting the sample to the sickest or some other subset would introduce spectrum bias, and the sample would end up looking more similar to the populations in which we already have copious data (ICU, OR/perioperative) than it would to the population of interest (ED patients with AHF and an undifferentiated broad spectrum of risk for serious outcomes). As a result, the information gained from a limited comparison like this would not be particularly novel compared to what we already cite from the ICU and OR literature, which in turn would make the ethics of performing such a comparison even less in favor of doing so (i.e. the information gained is less, but the risks to those patients receiving right heart catheterizations by emergency physicians, not trained to perform this procedure, would remain just as high). Moreover, doing so would effectively be besides the point: restricting an ED-based comparison to only those patients in whom invasive hemodynamic monitoring is indicated (i.e. the sickest patients) would directly ignore the patients for whom improved risk-stratification may be a benefit (i.e. those ED patients with AHF who may actually be low risk, but who will be admitted to the hospital anyway, because ED clinicians lack reliable tools for risk-stratification and tend to be risk-averse as cited in the introduction). Again, this would also hinge on the premise that right heart catheterizations were able to be performed by ED clinicians, which currently it is not (as part of their training or their clinical scope of practice, as cited). Thus, like the ECG, the point is less about purity of measure compared to an invasive gold standard, and more about reliability of that measure and its ability to be used as a screening tool to help risk-stratify patients at risk for patient-oriented outcomes.

Once again, given the complexity of the above (and the length of our response here, for which we apologize), we could not capture everything in our edits to the manuscript. Please let us know if further edits are needed, and outweigh the reviewer’s request for more brevity, and if so we are obviously happy to oblige.

- the control cohort of patients with sepsis might not be optimal and an additional comparison with healthy individuals would be appreciated

Thank you for this feedback. We chose the septic cohort (CC) as a control for three primary reasons: 1. These patients were enrolled at the same time, same sites, and the same parent study, as the AHF patients in the derivation cohort (DC) 2. Sepsis is fairly well defined to have different hemodynamic derangements, particularly with cardiac index and SVRI, compared to AHF. 3. Like AHF (as you point out in the next comment) Sepsis is also a clinical diagnosis (i.e. it is a clinical syndrome). Given these, it was felt that the CC represented a reasonable external control for which there should be minimal differences at the level of study/site/time, and for which the primary discernable difference should be hemodynamics and clinical diagnosis

However, we certainly agree that perhaps another control cohort may add value. Pursuant to your comment below (about AHF as a clinical diagnosis, and the inherent risk of conflating non-cardiac dyspnea with AHF even in an adjudicated study such as ours) and in light of the comment here, we performed an additional sensitivity analysis with a second control group perhaps more optimal than the septic patients. This new, second control cohort (“CC2” in the amended text, now in Supplement S4 to accommodate the reviewer request for more brevity in the methods), includes all patients enrolled in the current study (meeting all inclusion and exclusion criteria) but who were ultimately excluded after adjudication as “Not-AHF”. Meeting the inclusion criteria (emergency physician suspicion of AHF and at least 1 of: 1. Dyspnea OR 2. BNP>300 OR 3. Edema on chest x-ray) but being adjudicated as “Not-AHF” means (with the same limitations/uncertainty to the diagnostic adjudication as included below and in all AHF research) these patients should mostly be those who had non-cardiac dyspnea and/or chronic heart failure that was not acutely decompensated. Given this, one would expect these patients would have different hemodynamics as well. 

Lines 171-172, 521-522, 529-531 and 656-679 include added text to address the changes requested and described above. In short, there was significant difference (p<0.001 by PERMANOVA) in cardiac index/SVRI profiling (i.e. difference in hemodynamic profile) between the VC and this CC2 (non-cardiac dyspnea and/or heart failure without compensation), similar to the difference between the VC and the septic CC. 

- the diagnosis of AHF is clinical, the authors should reinforce (or acknwoledge in limitation) the criteria for inclusion as the risk of including patients with non-cardiac dyspnea is not minimal

Thank you for pointing this out. As discussed in the responses to reviewer 1, having a diagnostically-adjudicated sample was an explicit methodological choice directly meant to improve upon what the prior study’s authors (Nowak et al 2017) identified as one of their biggest limitations. However, we completely agree that even this improvement in rigor compared to the Nowak study is not without bias, as all AHF studies ultimately suffer from the limitation of AHF being a clinical diagnosis. We have added text at lines 656-679 to further clarify this limitation. For more on differentiation from non-cardiac dyspnea, see our response to your feedback above regarding an additional control cohort beyond the sepsis patients.

- the text is too long, in particular the introduction and methodological section, and sohuld be shortened and made clearer for clinical readers

Thank you for your feedback. We have moved the methods for the 3 sensitivity analyses to the Supplemental material (S4). Text has been deleted and/or amended for brevity and clarity in the introduction. 

We agree, it is a quite long and complex manuscript. We have tried to balance the desire for brevity and clarity with the need to accommodate the other requested edits and maintain methodological transparency, as discussed throughout the other responses to reviewer comments. Please let us know if further brevity is required, and we can move more text to the Supplement as needed and/or provide further edits.

---

## [Decision Letter · Decision Letter 1]

10 Mar 2022

Hemodynamic profiles by non-invasive monitoring of cardiac index and vascular tone in acute heart failure patients in the emergency department: external validation and clinical outcomes

PONE-D-21-36949R1

Dear Dr. Harrison,

We’re pleased to inform you that your manuscript has been judged scientifically suitable for publication and will be formally accepted for publication once it meets all outstanding technical requirements.

Kind regards,

Gianluigi Savarese

Academic Editor

PLOS ONE

Additional Editor Comments (optional):

Reviewers' comments:

Reviewer's Responses to Questions

**Comments to the Author**

1. If the authors have adequately addressed your comments raised in a previous round of review and you feel that this manuscript is now acceptable for publication, you may indicate that here to bypass the “Comments to the Author” section, enter your conflict of interest statement in the “Confidential to Editor” section, and submit your "Accept" recommendation.

Reviewer #1: All comments have been addressed

Reviewer #2: All comments have been addressed

2. Is the manuscript technically sound, and do the data support the conclusions?

Reviewer #1: Yes

Reviewer #2: Yes

3. Has the statistical analysis been performed appropriately and rigorously? 

Reviewer #1: Yes

Reviewer #2: Yes

4. Have the authors made all data underlying the findings in their manuscript fully available?

Reviewer #1: Yes

Reviewer #2: Yes

5. Is the manuscript presented in an intelligible fashion and written in standard English?

Reviewer #1: Yes

Reviewer #2: Yes

6. Review Comments to the Author

Reviewer #1: Thank you for your answers and clarification. I think that the manuscript has improved and in my opinion can be accepted for pulblication.

Reviewer #2: All the comments have been addressed by the authors. However, the lack of a validation cohort with invasive hemodynamic assessment and the lack of a control healthy cohort remain strong limitations that limit the routine applicability of the technique in clinical practice

7. PLOS authors have the option to publish the peer review history of their article (what does this mean?). If published, this will include your full peer review and any attached files.

Reviewer #1: No

Reviewer #2: No

---

## [Editor Report · Acceptance letter]

22 Mar 2022

PONE-D-21-36949R1 

Hemodynamic profiles by non-invasive monitoring of cardiac index and vascular tone in acute heart failure patients in the emergency department: external validation and clinical outcomes 

Dear Dr. Harrison:

I'm pleased to inform you that your manuscript has been deemed suitable for publication in PLOS ONE. Congratulations! Your manuscript is now with our production department. 

Kind regards, 

on behalf of

Dr. Gianluigi Savarese 

Academic Editor

PLOS ONE